# Balanced Ranking with Relative Centrality: A multi-core periphery perspective

**Chandra Sekhar Mukherjee** [*]
Thomas Lord Department of Computer Science
University of Southern California
chandrasekhar.mukherjee07@gmail.com

**Jiapeng Zhang** [†]
Thomas Lord Department of Computer Science
University of Southern California
jiapengz@usc.edu

## Abstract

Ranking of vertices in a graph for different objectives is one of the most fundamental tasks in computer science. It is known that traditional ranking algorithms can generate unbalanced ranking when the graph has underlying communities, resulting in loss of information, polarised opinions, and reduced diversity (Celis, Straszak & Vishnoi [ICALP 2018]).

In this paper, we focus on *unsupervised ranking* on graphs and observe that popular centrality-measure-based ranking algorithms such as PageRank may often generate unbalanced ranking here as well. We address this issue by coining a new approach, which we term *relative centrality*. Our approach is based on an iterative graph-dependent local normalization of the centrality score, which promotes balancedness while maintaining the validity of the ranking.

We further quantify the reasons behind this unbalancedness of centrality measures. using novel structure that we propose. We term this as the multi-core-periphery with communities (MCPC) structure. We provide theoretical and extensive simulation support for our approach towards resolving the unbalancedness in MCPC.

Finally, we consider graph embeddings of 11 single-cell datasets. We observe that top-ranked as per existing centrality measures are better separable into the ground truth communities. However, due to the unbalanced ranking, the top nodes often do not contain points from some communities. Here, our relative-centrality-based approach generates a ranking that provides a similar improvement in clusterability while providing significantly higher balancedness.

## 1 Introduction

Ranking of data points is one of the fundamental problems in computer science, with applications ranging from information retrieval (Liu et al., 2009), recommendation systems (Page et al., 1998), community detection (Liu et al., 2016; Hajij et al., 2020), resource allocation problems (Zipkin, 1980) and is also extensively used in online economies (Singh & Joachims, 2019).

Ranking algorithms are often applied to data where the data has an underlying partition into groups/ communities, such as when allocating resource (Celis et al., 2017), using ranking algorithms for community detection (Liu et al., 2016), or deciding influential nodes in social networks (where people of different ethnic backgrounds or political beliefs can be considered as communities) (Zhang et al., 2015; Kojaku & Masuda, 2017). Here in many cases it is desirable to have a *balanced ranking*, i.e., top data points ranked by the algorithm should contain a reasonable fraction data points from each community. While there has been recent progress in the supervised setting with respect to ranking with group fairness constraints (Celis et al., 2017; Singh & Joachims, 2019; Zehlike et al., 2022), the balancedness of ranking in the unsupervised learning context remains under-explored.

In this paper, we focus on unsupervised ranking on graphs with underlying communities. We note that the primary class of ranking algorithms here are Centrality measures. Different centrality measures

---

[*]Research supported by NSF CAREER award 2141536.
[†]Research supported by NSF CAREER award 2141536.

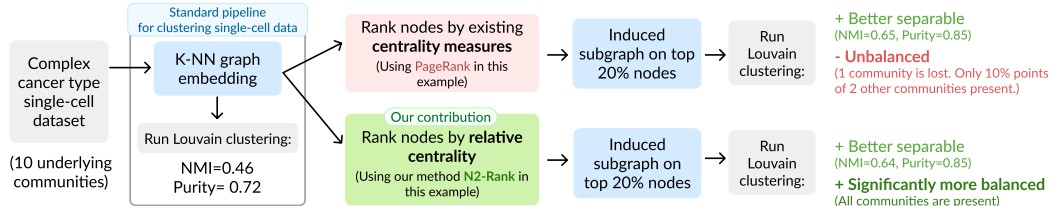

Figure 1: An example of the benefit of unsupervised balanced ranking algorithms. While the top-ranked nodes by existing centrality measures have improved separability (the induced subgraph can be clustered with better accuracy), the unbalancedness results in some communities being completely overlooked. In contrast, our relative-centrality-based algorithm offers significantly higher balancedness while having comparable separability.

rank the nodes of a graph according to their notion of importance. Centrality measures like PageRank (Page et al., 1998) have been hugely influential, and others such as Betweenness (Barthelemy, 2004), Closeness (Okamoto et al., 2008), Degree centrality (Zhang & Luo, 2017) and K-core ranking (Hébert-Dufresne et al., 2016) have been used in understanding phenomena such as cognitive behavior (Bassett et al., 2013), online amplification (Barberá et al., 2015), and others.

**Single-cell RNA seq data: Motivation and a use-case of unsupervised balanced ranking.**

Balancedness can be very important in the unsupervised context as well. As a concrete example, we consider single-cell RNA seq data, which is one of the most influential biological data of the last decade (Chattopadhyay et al., 2014; Poulin et al., 2016; Zhang et al., 2021; Li et al., 2024).

Here, we are given a dataset with some $n$ data points (each data point corresponding to a single cell), with $d$ features (each feature corresponding to the gene expressions of a cell). In single-cell analysis, the main goal is to understand cell behavior (understanding biological systems, diseases, and others) through gene expression.

Here, separating the data points into different clusters (according to their cell types) using gene expression is an important step in single-cell analysis, as noted by this popular Nature review paper (Heumos et al., 2023). Once the different communities are found, bioinformaticians then use it for different downstream tasks.

However, despite significant research and progress, the performance of the state-of-the-art community detection algorithm can be very poor (e.g., NMI less than 0.6). In such a case, centrality measures may be helpful. For example, we observe that if we run popular centrality measures like PageRank on the graph and choose to run community detection on only the induced subgraph of the top-ranked vertices, the performance of the community detection algorithm is much improved. However, the top-ranked points can be unbalanced, and some communities may be completely missed while others are underrepresented, which affects the downstream tasks. See Figure 1 for a detailed example. Motivated by this example, we focus on directed graphs. Our contributions are as follows.

## 1.1 CONTRIBUTIONS

**A Balanced meta-ranking-algorithm.** As the primary contribution, we coin a novel concept, "relative centrality", and design a meta ranking algorithm (Details in Section 3.1) that provides *superior balancedness* to several popular centrality measures on the graph embeddings of a large set of biological (single-cell) datasets.

**A new structural assumption.** Structural assumptions are ubiquitous in the design and analysis of unsupervised learning algorithms, used for both inspiration of new algorithms as well as analysis of established algorithms. In the context of ranking on graphs with underlying community structures, two such structures are community structure and core-periphery structure. The former is used for understanding community detection, and the latter is heavily interlinked with the performance of centrality measures. We refer the readers to Figure 2 for a schematic representation of these two structures. Here, the recent review (Yanchenko & Sengupta, 2023) on core-periphery structures has noted *"a better understanding of the interplay between community and core-periphery structure"* as an interesting open problem.

Against this backdrop, we propose a new structure that we call a *multi-core-periphery with communities* (MCPC) by combining these two structures to quantify the unbalancedness in traditional centrality measures observed in real-world graphs. Readers may refer to Figure 3 to gain an initial overview and refer to Section 2 to get a more detailed insight into this structure. We provide theoretical proof of the unbalancedness in random graphs with MCPC structure and show that our relative centrality approach overcomes this, demonstrating initial theoretical evidence and large-scale simulation support.

**Applications to real-world data in improving inference of community structure.** Finally, as a concrete application of our balanced ranking algorithm, we show that if we apply graph clustering algorithms to the induced subgraph of top-ranked nodes of the aforementioned datasets, we have a comparable improvement in clustering accuracy to existing centrality measures, while our result is *significantly more balanced*. Details can be found in Section 4.

We end this section by laying out a detailed organization of the rest of the paper.

**Organization.**   In Section 2, we provide further background on the community and core-periphery structures and the motivation behind our multi-core-periphery structure, and then formalize the MCPC structure and the characteristics of an ideal ranking algorithm in this framework. We further discuss connections to real world data and other models.

Then, in Section 3, we define a random graph model in Section as an instantiation of MCPC and analyze the unbalancedness in traditional centrality measures. In Section 3.1, we define our relative centrality approach and describe our algorithms, along with initial theoretical and simulation support for its balancedness. We also evaluate these phenomena on graph embeddings of a generalized mixture model in Appendix C.

Section 4 contains our experiments on the real-world graphs. Motivated by the standard clustering pipeline of single-cell data, we look at the graph embeddings of 11 complex single-cell datasets. For all the datasets, selecting top-ranked nodes by centrality measures leads to improved separability (observed both structurally and through improved performance of clustering algorithms), with our methods providing significantly superior balancedness compared to traditional centrality measures.

Finally, while we make exciting progress in balanced unsupervised ranking algorithms and show both theoretical analysis and impact on real-world datasets, we note that our work has some limitations. We place a detailed discussion in Appendix E.3.

We conclude this Section with a discussion of related works.

## 1.2   COMPARISON WITH EXISTING WORK.

In this paper, we design a balanced ranking algorithm that is applicable to a large set of real-world data and can also be theoretically supported in the MCPC structure. Even within the core-periphery literature, work with multiple cores is rarer. The existing works here mainly focused on comparably restrictive settings. The works (Tunç & Verma, 2015) and (Kojaku & Masuda, 2017) proposed specific equations to capture the interaction between nodes from different cores and designed maximum likelihood-based algorithms for these equations; the paper Elliott et al. (2020) studied directed graphs with multi-cores, but they do not consider the coexistence of community structure. As such, we applied the algorithms of Elliott et al. (2020) to our graph simulation model and observed an almost-random outcome and thus placed it in Appendix E, along with a more in-depth comparison with the literature as well as some limitations of our work. Most importantly, to the best of our knowledge, none of the previous papers considered the unbalancedness issue of centrality measures in a similar setting.

## 2   MULTI CORE-PERIPHERY WITH COMMUNITIES (MCPC): A THEORETICAL LENS FOR CENTRALITY MEASURES IN GRAPHS WITH COMMUNITIES

In this section, we lay out the intuition behind the MCPC structure to capture the unbalancedness in the ranking due to traditional centrality measures, starting by discussing two popular structures.

**Community structure.** A graph is thought to have community structure if the vertices have an underlying (unknown) partition into some $k$ communities such that vertices from the same underlying community are more likely to be connected by edges than vertices from different communities.

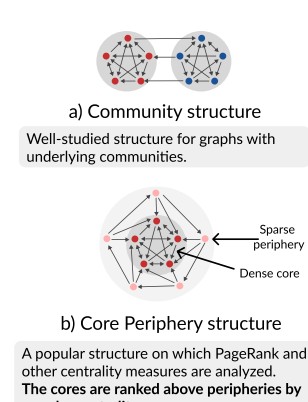

a) Community structure

Well-studied structure for graphs with underlying communities.

Sparse periphery

Dense core

b) Core Periphery structure

A popular structure on which PageRank and other centrality measures are analyzed. **The cores are ranked above peripheries by popular centrality measures.**

Figure 2: Structures in analysis and design of centrality measures and community inference algorithms

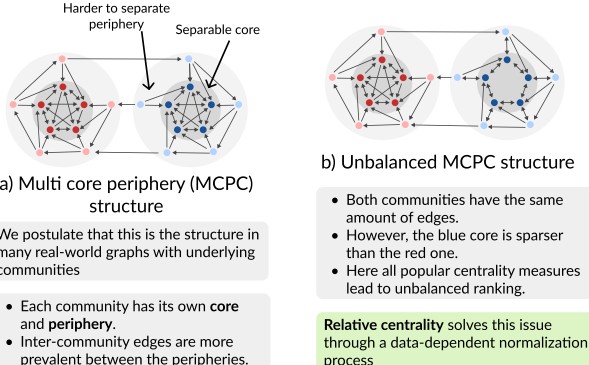

Harder to separate periphery    Separable core

a) Multi core periphery (MCPC) structure

We postulate that this is the structure in many real-world graphs with underlying communities

- Each community has its own **core** and **periphery**.
- Inter-community edges are more prevalent between the peripheries.

b) Unbalanced MCPC structure

- Both communities have the same amount of edges.
- However, the blue core is sparser than the red one.
- Here all popular centrality measures lead to unbalanced ranking.

**Relative centrality** solves this issue through a data-dependent normalization process

Figure 3: Example of graphs with MCPC structure. Unlike the graphs with community structure where inter-community edges are spread uniformly in a community, here, the (dense) cores have a smaller fraction of inter-community edges. In (a), both cores have the same density (they are identical) and centrality measures such as degree centrality will result in balanced ranking. In (b) the blue core is sparser, and therefore existing centrality measures will rank them lower.

This structural assumption has inspired important clustering objectives such as modularity (Brandes et al., 2007; 2006; Noack, 2009) and correlation clustering (Bansal et al., 2004) and also popular probabilistic models such as the stochastic block model (SBM) (Abbe, 2018; Mukherjee et al., 2024). Figure 2 (a) gives a schematic representation of a graph with community structure.

**Core periphery (CP) structure.** We now turn our focus on a different kind of structure called core-periphery (CP) structure which has been extensively studied in the network analysis literature in the last two decades (see the surveys (Rombach et al., 2014; 2017; Yanchenko & Sengupta, 2023) and the references therein). Here, the graph is assumed to contain a dense subgraph (the core), and the other vertices (forming the periphery) are sparsely connected to each other, as well as the core. Figure 2 (b) gives a schematic representation of a graph with core-periphery structure.

**Centrality measures perform well on core-periphery structure.** Identifying the cores is an important task in graphs with core-periphery structure (Cucuringu et al., 2016; Rombach et al., 2014; Yanchenko & Sengupta, 2023). Here centrality measures (such as degree centrality and PageRank) are known to be very good at ranking the core nodes above periphery nodes (Rombach et al., 2014; Yanchenko & Sengupta, 2023) even being near-optimal in some random graph setting (Barucca et al., 2016), making CP structures a promising setup to better understand balancedness of centrality measures in graphs with underlying community structure.

However, the majority of the work in the core-periphery structure is in a single-core setting. Some recent papers have observed the existence of multiple cores in graphs (Wang et al., 2011; Elliott et al., 2020; Kojaku & Masuda, 2017), and also coexistence of CP and community structure (Tunç & Verma, 2015; Yang & Leskovec, 2014), with the recent survey (Yanchenko & Sengupta, 2023) noting *"a better understanding of the interplay between community and CP structure"* as an open problem. To the best of our knowledge, none of the existing works discussed the balancedness of centrality measures in the presence of multiple cores. Here we propose the following structure.

**The MCPC structure.** Consider a graph $G(V, E)$ with a hidden partition into communities $V_1, \ldots, V_z$. We say that the graph satisfies an MCPC-structure w.r.t. the hidden partition if: each set $V_i$ has a further partition into a *core* $V_{i,1}$ and *periphery* $V_{i,0}$ such that most edges starting in a core end up in that core itself (the cores are *densely connected*). The peripheries are *loosely connected*. We formalize this notion with **core concentration** as follows.

**Definition 2.1** (Core concentration). Given a directed graph $G(V, E)$, for any $V', V'' \subseteq V$, let $E(V', V'')$ denote the number of edges starting in $V'$ and ending in $V''$. Then we define the core concentration (or simply concentration) of $S \subseteq V$ as

$$\mathsf{CC}_G(S) = \big(E(\bar{S}, S) - E(S, \bar{S})\big)/|E(S, V)| \tag{1}$$

That is, we expect the core to have two properties. First, only a few edges originating in a core should leave the core, which we penalize with the $-E(S, \bar{S})$ term. Secondly, we expect more edges from the peripheries ending at the cores (making each core a more central part of the corresponding community), which we incentivize with the $+E(\bar{S}, S)$ term.

Next, we expect the cores to have *fewer inter-community edges* compared to the peripheries. Note that if such a situation were to occur, the top-ranked core-ranking algorithms will be easier to separate, capturing the behavior of PageRank in our initiating example in Figure 1. Then, the concrete definition of the MCPC structure is as follows.

**Definition 2.2.** We say a graph $G(V, E)$ with an underlying hidden partition $\{V_1, \ldots, V_z\}$ has an $(\alpha, \beta)$-MCPC structure for $\alpha = \Omega(1) > 0$ and $\beta < 1$ iff

i) The hidden partition imposes a community structure on the graph, i.e., $E(V_i, V_i) \geq E(V_i, \bar{V}_i)$.

ii) Each hidden partition $V_i$ can be further partitioned into two sets, namely the core $V_{i,1}$ and the periphery $V_{i,0}$, such that $\mathsf{CC}_G(V_{i,1}) > \mathsf{CC}_G(V_{i,0}) + \alpha$. (the core $V_{i,1}$ has a higher core concentration than periphery $V_{i,0}$).

iii) For any two communities $V_i, V_j$, $\frac{E(V_{i,1}, V_j)}{|V_{i,1}|} \leq \beta \frac{E(V_i, V_j)}{|V_i|}$. (There are fewer inter-community edges between cores than the peripheries).

In this context, in Figure 3 (a), the two cores have almost identical concentration, whereas the peripheries have a lower concentration than the cores. Therefore, a simple degree centrality (ranking the nodes in descending order of their degree) will rank the cores above peripheries, while also ranking the cores in a balanced manner.

**Unbalancedness of Centrality measures on graphs with MCPC structure.** However, the balancedness of centrality measures may deteriorate quite fast if the different cores have different levels of connectedness. For example, let us consider the instance in Figure 3(b). Concretely, it is a 3-regular directed graph (i.e., each node has the same out-degree, and both the communities (blue and red) have the same number of core and periphery vertices (5 each)). However, the blue core has a lower core-concentration than the red core. In such a case, degree centrality will result in an *biased/unbalanced* ranking of the nodes. Note that the downstream algorithm will not know the total number of cores or core vertices. In such a case, if one selects points from the top of the ranking, it may lead to an unbalanced representation of the communities (as we observed in the example of the cancer dataset). Against this backdrop we define the two primary characteristics we want in a ranking algorithm.

**Definition 2.3** (Performance metrics of a ranking algorithm w.r.t. the MCPC structure.)**.**

i) *Core-prioritization:* A ranking algorithm should rank core vertices above periphery vertices. We quantify this as the AUROC value (Marques et al., 2023) of the w.r.t. core/periphery label of each vertex, which is close to 1 when almost all core vertices are placed above almost all periphery vertices.

ii) *Balancedness:* Let $F$ be a ranking of the vertices, and let $F_c$ denote the top $c \cdot n$ vertices in the ranking. Then, we quantify the balancedness in the top $c \cdot n$ vertices in the ranking as $\mathcal{B}_c(F, G) := \left(\min_i \frac{|F_c \cap V_{i,1}|}{|V_{i,1}|}\right) / \left(\max_i \frac{|F_c \cap V_{i,1}|}{|V_{i,1}|}\right)$. That is, the top points in the ranking should contain a roughly equal proportion of points of each core. The ranking reaches a perfect balancedness if $\mathcal{B}_c(F, G) = 1$. The **total balancedness** is defined as $\mathcal{B}(F, G) := \frac{1}{n} \sum_{i=1}^{n} \mathcal{B}_{i/n}(F, G)$.

**Relative centrality mitigates the unbalancedness.** We show in Section 3 that the relative centrality measure we propose achieves a balanced ranking with high core-prioritization even when the cores have different concentrations. This provides a theoretical understanding of the unbalancedness of existing centrality measures and evidence of our algorithm's performance in a controlled (yet important) setting. We further present extensive experiments in Section 4.1 to demonstrate the usefulness and balancedness of our algorithm on real-world data.

MCPC **in the real world; Single-cell data and beyond.** First, we give some more insight on why single-cell data is a good fit for this structure. These datasets can be considered as high-dimensional (usually more than 25,000) noisy data, suffering from technical noise and experimental error Jovic et al. (2022). Many data points could be abnormal cells, such as dead cells and doublets, or cells affected adversely by experimental noise. Therefore, separating such cells in the graph embeddings

may be very hard. In our MCPC structure, we capture them as peripheries. In contrast, the cells with less degenerative effect can be seen as cores. This further motivates the design and analysis of balanced ranking algorithms, as one would hope to select cores for each cell type. In Section 4.1, we observe that the communities in graph embeddings of real-world biological data indeed contain cores with higher concentration values, and these cores have a lower fraction of *inter-community edges* compared to the whole data.

Beyond this application, MCPC structure may also be relevant to certain social choice scenarios. For example, if we consider the social network of US population through the lens of political affiliation, the resultant data should exhibit some MCPC structure. This is because there will be people who are firmly committed to one political affiliation, either Democrats or Republicans; these people will form the core of each community. In contrast, more neutral people will align with different affiliations for different topics, forming the peripheries. Some recent papers have observed that social networks such as Twitter can have multiple cores (Kojaku & Masuda, 2017; Yang et al., 2018).

## 3 ANALYSIS AND EXPLORATION OF CENTRALITY AND RELATIVE CENTRALITY IN MCPC VIA A RANDOM GRAPH MODEL

A general graph with an MCPC structure may be too complicated to both simulate and analyze. In this direction, we define a block model-based random graph setup to instantiate graphs with MCPC structure. In a block model, the set of vertices $V$ are divided into some blocks, and then the nature of interaction between two vertices is a function of their block identity. For example, the popular stochastic block model (Abbe, 2018) (SBM) is such a model, used to generate graphs with underlying community structure. Block models are also popular in CP literature, with the performance of many algorithms made under different CP assumptions quantified on these block model-generated graphs (Cucuringu et al., 2016; Elliott et al., 2020; Zhang et al., 2015; Yang et al., 2018).

We take a similar approach in this paper and define a model that we call MCPC-block model that generates a directed graph where the out-degree of each vertex is $k(1 \pm o(1))$ for some $k$. For simplicity, we define it w.r.t. $\ell = 2$ underlying community, which can be easily extended.

**Definition 3.1** (MCPC-block model). We have $V = \{v_1, \ldots, v_n\}$, a set of $n$ vertices that are partitioned into 2 communities $V_1$ and $V_2$, with each $V_i$ further partitioned into a core $V_{i,1}$ and a periphery $V_{i,0}$. There is a $4 \times 4$ block-probability matrix $\mathbb{P}$ such that each row sums to 1. Then, for any $v_i \in V_{\ell,c}$ and $v_j \in V_{\ell',c'}$, we add an $v_i \to v_j$ edge iid with probability $k/|V_{\ell',c'}| \cdot \mathbb{P}[(\ell, c), (\ell', c')]$.

**Baseline centrality measures.** We consider degree centrality, PageRank (with 3 different damping factors), Katz Centrality (Bloch et al., 2023), and a popular core-decomposition-based algorithm (that we call onion decomposition) (Hébert-Dufresne et al., 2016) as initial baseline algorithms for evaluation of centrality measures on MCPC structures.

**Centrality measures can be unbalanced even when both communities and cores are of same size.** We set $|V_{\ell,c}| = 0.25 \times n$ for all the blocks and then set the block parameters to lead to an $(\alpha, \beta) - $ MCPC structure. Here, it is important to note that while we demonstrate the phenomenon in theory and in simulation when the size of all the cores and peripheries are the same, our algorithms are applicable even when they are of different sizes. In fact, in our real-world experiments, the size of the different communities in a dataset varies widely. Then, the behavior of degree centrality can be captured as follows.

**Theorem 3.2** (Behavior of degree centrality). Let $G(V, E)$ be a graph sampled from the MCPC-block model w.r.t. partition of $V$ into $V_{\ell,c}, (\ell, c) \in \{0, 1\}^2$ where $k = \omega(\log n)$. Let $F(v)$ be the degree of the vertex. Then for any $v_i \in V_{\ell,1}$ we have $F(v_i) = 2k + k \cdot (1 \pm o(1))(\mathsf{CC}_G(V_{\ell,1}))$.

That is , the degree of a vertex is almost linearly related to the $\mathsf{CC}_G$ of the core it belongs in. If $\mathsf{CC}_G(V_{\ell,1}) > \mathsf{CC}_G(V_{\ell',1}) + C$ for any constant $C$, the degree of all vertices in $V_{\ell,1}$ will be higher than the ones in $V_{\ell',1}$. This will result in an *unbalanced* ranking. The proof of Theorem 3.2 can be found in Appendix A.1. Next, we observe this in simulation.

**Initial simulation.** For simulation purposes, we instantiate $\mathbb{P}$ as in Table 1, parameterized with $\gamma$. When $\gamma = 0$, the generated graph will have $\alpha \approx 0.3$ and $\beta \approx 0.25$, and $\mathsf{CC}_G(V_{0,1}) \approx \mathsf{CC}_G(V_{1,1})$. In such a case, degree, as well as the other centrality measures, have high balancedness throughout the ranking, as observed in Figure 4(a). Moreover, it has a high core prioritization (as the concentration

(a) ICEF of induced (b) Balancedness of (c) ICEF of induced sub-(d) Balancedness of rankings for $\gamma =$ subgraph for graph gen- rankings in a graph graph $\gamma = 0.05$ 0.05 erated with $\gamma = 0$ generated from $\gamma = 0$

Figure 4: Balancedness and intra-community edge fraction (ICEF) of the induced subgraph of the top-ranked nodes due to different centrality methods and our initial method (N-Rank) on 2-block MCPC graphs generated with parameter $\gamma$ as per Table 1. When $\gamma = 0$, in the generated graph both the cores have roughly equal concentration and the centrality measures produce a balanced ranking. When $\gamma > 0$, while the ICEF still increases at the top of the ranking (indicating better separability), the unbalancedness increases, which is mitigated by our method.

of peripheries is lower). Thus, the induced graph of the top $c$-fraction of the points has a *higher intra-community edge fraction (ICEF)* (which is simply the fraction of edges with endpoints being in the same community) as $c$ decreases, which we note in Figure 4(b).

**Unbalancedness when cores have varying concentration.** However, the scenario changes if $\gamma > 0$. Consider $\gamma = 0.05$. Then we have $\mathsf{CC}_G(V_{1,1}) \approx \mathsf{CC}_G(V_{0,1}) + 0.07$. While the core prioritization will still be high for the centrality measures, the balancedness becomes very low, as shown in Figure 4(c).

### 3.1 MITIGATING UNBALANCEDNESS WITH RELATIVE CENTRALITY FRAMEWORK

The primary reason behind the unbalancedness of the traditional centrality measures is that they capture the global centrality of vertices. With this observation, we aim to develop a method that generates a scoring with the following properties.

i) The core vertices should be assigned a higher score than the periphery vertices.

ii) The score of two vertices belonging to different cores should be similar, *irrespective* of the cores they belong to.

If a scoring of the vertices satisfies the two properties, we call it a *relative centrality* measure. Then ranking vertices in descending order according to some score should lead to high balancedness as well as core prioritization. We first propose an initial Algorithm 1, that we call N-Rank.

---

Algorithm 1: NeighborRank (N-Rank) with $t$-step initialization

**Input:** Graph $G(V, E), t$. Let the Adjacency matrix be $A$.
$\mathbf{s} \leftarrow \mathbf{1}_n$                                        {#Vector of all ones}
**for** i in 1:t **do**
    $\mathbf{s} \leftarrow A^T \mathbf{s}$                                {# Obtaining initial centrality score}
**end for**
$F^{(t)}(v_i) \leftarrow \mathbf{s}[i]$
**for** $v_i \in V$ **do**
    $S_{v_i} \leftarrow \{v_j : v_j \in N_G(v_i), F^{(t)}(v_j) > F^{(t)}(v_i)\} \cup \{v_i\}.$    $\begin{Bmatrix} N_G(v_i) \text{ is the neighborhood of} \\ v_i \text{ based on the outgoing edges} \end{Bmatrix}$
    $\hat{F}^{(t)}(v_i) \leftarrow \dfrac{F^{(t)}(v_i)}{\underset{v_j \in S_{v_i}}{\text{average}}[F^{(t)}(v_j)]}$    { The relative-centrality step}
**end for**
**return** $\hat{F}^{(t)}$

---

#### 3.1.1 DESCRIPTION OF OUR FIRST METHOD N-RANK AND ANALYSIS IN THE TWO BLOCK MODEL

As discussed, we want an "initial centrality measure" that ranks the core vertices above periphery vertices (so far, we are independent of the balancedness).

We define a generalized class of measure $F^{(t)}$ via a $t$-step power method, as defined in Algorithm 1. When $t = 1$, this converges to the in-degree centrality measure. As we mostly focus on (almost)-regular directed graphs, this is essentially the same as the degree centrality measure. When $t > 1$, it (approximately) captures the $t$-step random-walk reachability of vertices. We note that we focus on (almost) regular directed graphs in this paper motivated by K-NN embeddings of real-world vector datasets. However, our algorithms can be applied to non-regular directed graphs as well.

Then, let us focus on the performance of the initial centrality measure with $t = 1$. Theorem 3.2 states that vertices from a core with higher $CC_G$ will get a higher score as per the initial centrality measure. To mitigate this unbalancedness, for each $v \in V$, we select the $v' \in N_G(v)$ ($v$'s neighborhood) with *higher $F$ score* and obtain the final score $\hat{F}(v)$ as the ratio of $F$ value of $v$ with average of these neighbors (including $v$ itself). Note that this value can be at most 1.

Consider any vertex $v \in V_{\ell,1}$. Furthermore, let $\beta$ be small ($o(1/k)$), i.e., inter-community edges between different cores are few. Then, for many such $v$, all of its neighbors will belong to either the same core, which will have a very similar score, or peripheries, which will have a lower score as they belong to a set with a lower concentration, following Theorem 3.2. Then, we have $\hat{F}(v) \approx 1$ for any $v \in V_{\ell,1}$, *irrespective* of which core it is. We capture this behavior in Theorem A.7 in Appendix A.2 where we show that almost core vertex has a $\hat{F}(v_i)$ value of $1 - o(1)$ with high probability. This generates initial intuition about the balancedness of N-Rank. Then, in Lemma A.8 of Appendix A.3, we show that the peripheries will have a lower score than the core vertices, which indicates 1-step N-Rank has a *high core prioritization*.

Finally, we show that on expectation, roughly the same fraction of points from each core has the score of $\hat{F}(v) = 1$, which is the maximum possible score. In contrast, following Theorem 3.2, the degree-centrality method will have points only from the stronger core at the top of the ranking.

**Theorem 3.3** (1-step N-Rank places points from each core at the top). Let $G$ be an $n$ vertex graph obtained from the MCPC block model with $k = \omega(\log n)$ resulting in an $(\Omega(1), o(1/k)) - \mathsf{MCPC}$-structure w.r.t. to the core-periphery blocks and $\hat{F}(v)$ be the score of the vertices as per Algorithm 1 for $t = 1$. Then, we get the following behavior on expectation for any core $V_{\ell,1}$.

$$\frac{(1 - o(1))|V_{\ell,1}|}{\mathbb{P}[(\ell,1),(\ell,1)] \cdot k} \leq \mathbb{E}[|V_{\ell,1} \cap \{v : \hat{F}(v) = 1\}|] \leq \frac{(1 + o(1))|V_{\ell,1}|}{\mathbb{P}[(\ell,1),(\ell,1)] \cdot k}$$

That is, on expectation, a similar fraction of points ($\Theta(1/k)$) from each core will have a score of 1 in the 1-step N-Rank method. This provides an initial theoretical insight into why the highest-ranked points in N-Rank should be balanced. The proof can be found in Appendix A.4.

In fact, we observe in all our simulations that N-Rank has a high balancedness while having similar ICEF improvement compared to the baselines (Figure 4(d)). Figure 4(c) shows that N-Rank has high balancedness for $\gamma = 0.05$, even though the two cores have different concentrations. Thus, N-Rank with 1-step can be thought of as a way to create a relatively central version of degree centrality.

**Generalization into a meta-algorithm.** Next, we generalize our algorithm in two natural ways.

1) There may be periphery vertices in the graph that have a high $F_G$ value compared to its 1-hop neighborhood (E.g., a periphery vertex that has no edges going to a core). To mitigate this issue, we can look at some $y$-hop neighborhood $N_{G,y}(v)$ of $v$ when selecting the reference set.

2) We have observed that our N-Rank approach increases the balancedness in the initial centrality measure $F_G$. In this direction, we can recursively apply this process by first calculating the $y$-hop N-Rank value and then feeding it back to the algorithm as the initial centrality measure to further increase balancedness. We can apply this recursive process any $z \geq 1$ many times.

Due to space restriction, we write the general method as a meta-algorithm M-Rank(t,y,z) 2 in Appendix B with an in-depth discussion. Then, N-Rank can be written as MR-Rank(1,1,0). We further define MR-Rank(t,2,0) as N2-Rank ($t$ steps) and MR-Rank(t,1,1) as RN-Rank ($t$ steps). M-Rank(t,y,z) has a runtime of $\mathcal{O}((t \cdot |E| + (z - 1) \cdot k^y \cdot |V|)$ for $k$-regular directed graphs.

**Large scale simulation and core-prioritization vs. balancedness tradeoffs.** We use the $\gamma$-parameterized block probability matrix in Table 1 and vary $\gamma$ from 0 to 0.2 with an increment of 0.02, generating a graph for each value for a large-scale simulation. We make the following observations.

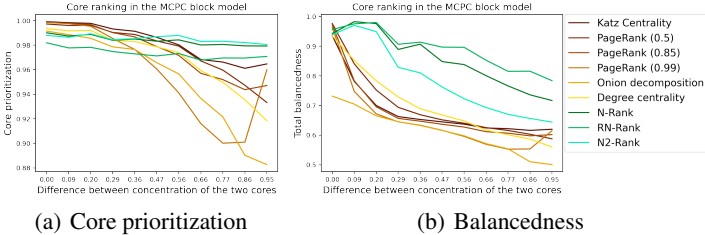

(a) Core prioritization                   (b) Balancedness

Figure 5: Core-prioritization and total balancedness (as per Definition 2.3) of centrality measures on graphs generated MCPC-block model for different $\gamma$ values (As per Table 1). As $\gamma$ increases, the difference in the concentration of the cores from the two communities increases, and as a result, the gap in the balancedness of our method becomes higher (while core-prioritization remains similar) compared to existing centrality measures.

As $\gamma$ increases, three structural changes occur in the graph. $\alpha$ (separation of CC values between cores and peripheries) decreases, $\beta$ (inter-core edge fraction) increases, and the difference between the core concentration values of the two cores becomes larger. We observe in Figures 5(a) and 5(b) that as this happens, our balancedness becomes superior to the baselines while maintaining comparable core prioritization.

|         | $V_{0,1}$ | $V_{0,0}$ | $V_{1,0}$ | $V_{1,1}$ |
|---------|-----------|-----------|-----------|-----------|
| $V_{0,1}$ | $0.8+\gamma$ | $0.075-\frac{\gamma}{4}$ | $0.075-\frac{\gamma}{4}$ | $0.05-\frac{\gamma}{2}$ |
| $V_{0,0}$ | $0.4$ | $0.2$ | $0.2$ | $0.2$ |
| $V_{1,0}$ | $0.2$ | $0.2$ | $0.2$ | $0.4$ |
| $V_{1,1}$ | $0.8+\gamma$ | $0.075-\frac{\gamma}{4}$ | $0.075-\frac{\gamma}{4}$ | $0.05-\frac{\gamma}{2}$ |

Table 1: Block probability matrix to generate MCPC-block model graph

## 4 EXPERIMENTS ON REAL-WORLD DATA

Motivated by the workflow of single-cell data, we focus on $k$-NN embedding of vector datasets. That is, given a vector dataset $X$ with $n$ datapoints $\vec{x_1}, \ldots, \vec{x_n}$, we generate a graph $G_X(V, E)$ with $|V| = n$ and for each datapoint $\vec{x_i}$, we add directed edges starting from $v_i$ and ending at the representative vertices of $k$-nearest neighbors.

In this direction, we explore the relative centrality measures in a more complex simulation model based on the Gaussian mixture model (Reynolds et al., 2009) in Appendix C. As before, our methods have higher balancedness when different cores have different concentration values.

### 4.1 SINGLE-CELL DATA

**Datasets:** We use a total of 11 datasets. We use the 7 datasets from a recent database (Abdelaal et al., 2019), the popular Zheng8eq dataset (Duò et al., 2018), and two more large datasets (Smith et al., 2019), and a T-cell dataset (Savas et al., 2018) of cancer patients. All of these datasets have annotated labels available of their corresponding cell types that form the underlying communities. The size of the datasets vary from $1400$ to $54,000$. For each dataset, we first pre-process it with a standard pipeline (details in Appendix D) and then obtain its 20-NN graph embedding, which we denote as $G_0$. This is very similar to the pipeline of the state-of-the-art clustering method for single-cell data, Seurat (Stuart et al., 2019). Then we apply the baseline centrality measures and our relative centrality-based algorithms to these graphs. We make the following observations.

**Inference.** For each graph $G_0(V, E)$, we select some top $c$-fraction of the points in the descending order of their ranking as per the ranking algorithm $F$ and denote it as $F_c(V)$. First, we explain the observations w.r.t. the T-cell dataset (Savas et al., 2018).

i) **Core-ranking improves community structure.** We observe that as $c$-decreases, the corresponding induced subgraph has a *higher intra-community edge fraction*, as observed in Figure 6(a).

ii) **Evidence of cores.** Let $S$ be the top $c$-fraction of the points. Then, we have the very interesting observation that $\mathsf{CC}_{G_0}(V_i \cap S) > CC_{G_0}(V_i)$. That is, the higher-ranked points indeed form the cores of their communities as per our Definition 2.2. We discuss this in detail in the Appendix D.

iii) **Relative centrality has a higher balancedness.** The single-cell datasets we look at have many ground truth communities (sub-populations), and we aim to keep points from all communities at the

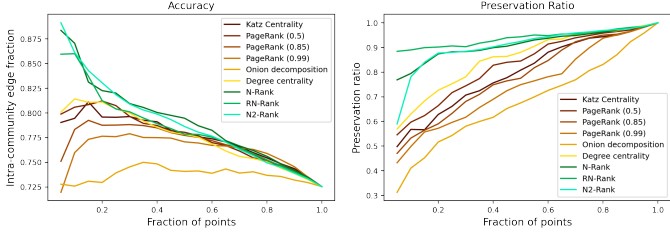

(a) ICEF of induced subgraph     (b) Preservation ratio of the subset

Figure 6: Improvement in intra-community edge fraction and balancedness for the top-ranked nodes of different centrality measures in the (Savas et al., 2018) dataset

top of the ranking. Thus, balancedness, as defined in Definition 2.3, is unsuitable, as it only captures the behavior of the worst-preserved community. In this direction, we define the following metric.

**Definition 4.1** (Preservation ratio). Given a set of points $V$ with an underlying partition $V_1, \ldots, V_z$ and a subset $S \subset V$, the preservation ratio of $V'$ w.r.t. the underlying partition is defined as $PR(V, V') = \frac{|V|}{z|S|} \cdot \sum_{i=1}^{z} \min \left\{ \frac{|S \cap V_i|}{|V \cap V_i|}, \frac{|S|}{|V|} \right\}$.

That is, each ground truth cluster contributes the minimum of $\frac{|V' \cap V_i|}{|V \cap V_i|}$, $\frac{|V'|}{|V|}$ to the term. We want to observe what fraction of points are necessary to achieve proportionality. The higher this value, the more clusters have a $|V'|/|V|$ fraction of the points in the filtered set. Note that the value of $PR(V, V')$ lies between $1/\ell$ and $1$. Furthermore, when there are only two communities, any set's preservation and balancedness values are related by a fixed linear equation. Then we observe that our methods have a *superior preservation ratio* throughout for the T-cell dataset in Figure 6(b), with RN-Rank having the best performance. In fact, at the $c = 0.2$, the baseline methods *completely miss* to include vertices from some communities, which we succeed at. The plots of all of the other datasets, along with a summarization, can be found in Appendix D.

**Improvement in clustering outcome.** Finally, we observe that not only does the induced graph by $F_c(V)$ have higher ICEF, but the subgraph is also better separable into its ground truth communities. To this end, we set $c = 0.2$ (the results are robust to the choice of the cutoff point). Then, we apply the well-known Louvain algorithm (Blondel et al., 2008) on the original 20-NN embedding as well as the induced subgraph for each CR algorithm. We compare the purity of the outcome on these points compared to the whole graph. We present the results on 7/11 datasets in Table 2. As before, our methods again demonstrate a superior preservation ratio, with *RN-Rank being the best*, while having comparable improvement in clustering accuracy. We provide the results for the other datasets, NMI improvement for top 20% in Appendix D. As with the experiments presented here, our methods provide a superior preservation ratio throughout the ranking for almost all of the datasets and have a similar improvement in ICEF as well as clustering outcome (NMI, purity) compared to the baselines. We add further discussion around the clustering improvement-balancedness tradeoff in Appendix E.

| Datasets | BH | | Se | | Tcell | | ALM | | AMB | | TM | | VISP | |
|---|---|---|---|---|---|---|---|---|---|---|---|---|---|---|
| # of points | 1886 | | 2133 | | 5759 | | 10068 | | 12832 | | 54865 | | 15413 | |
| Metrics | PR | Purity | PR | Purity | PR | Purity | PR | Purity | PR | Purity | PR | Purity | PR | Purity |
| Original values | 1.00 | 0.93 | 1.00 | 0.89 | 1.00 | 0.72 | 1.00 | 0.44 | 1.0. | 0.46 | 1.00 | 0.86 | 1.00 | 0.48 |
| Katz | 0.48 | 1.00 | 0.50 | 1.00 | 0.60 | **0.85** | 0.43 | 0.73 | 0.51 | 0.78 | 0.74 | 0.98 | 0.49 | 0.73 |
| PageRank (0.5) | 0.51 | 1.00 | 0.50 | 1.00 | 0.69 | 0.84 | 0.47 | 0.72 | 0.57 | 0.76 | 0.76 | 0.98 | 0.53 | 0.71 |
| PageRank (0.85) | 0.49 | 1.00 | 0.51 | 1.00 | 0.60 | 0.84 | 0.42 | 0.74 | 0.49 | 0.79 | 0.73 | 0.98 | 0.47 | 0.72 |
| PageRank (0.99) | 0.45 | 1.00 | 0.50 | 1.00 | 0.57 | 0.85 | 0.40 | **0.76** | 0.44 | 0.80 | 0.72 | 0.99 | 0.45 | 0.73 |
| Degree | 0.52 | 1.00 | 0.50 | 1.00 | 0.73 | 0.81 | 0.52 | 0.69 | 0.61 | 0.71 | 0.78 | 0.96 | 0.57 | 0.67 |
| Onion | 0.34 | 1.00 | 0.22 | 0.98 | 0.50 | 0.77 | 0.24 | 0.73 | 0.35 | **0.82** | 0.51 | **0.97** | 0.35 | **0.75** |
| RN-Rank | **0.68** | 0.99 | **0.56** | 0.99 | **0.89** | 0.79 | **0.61** | 0.71 | **0.69** | 0.69 | **0.87** | 0.95 | **0.61** | 0.68 |
| N2-Rank | 0.50 | 1.00 | 0.52 | 1.00 | 0.87 | **0.85** | 0.51 | 0.71 | 0.63 | 0.73 | 0.85 | 0.96 | 0.56 | 0.69 |

Table 2: Preservation ratio of top 20% points and purity score of Louvain on the induced subgraph (compared to the entire graph) for graph embeddings of single-cell data.

**Ethics statement**    In this paper we have focused on balanced ranking, with our experiments focused on single-cell RNA seq data. As such, we do not see any direct ethical concern with our current work. However, balanced ranking, in general, has the potential to reduce bias in different contexts, and future work in this direction should be ethically verified.

**Reproducibility statement**    Our paper consists of theoretical proofs on the MCPC structure, simulations on this structure, as well as large-scale real-world experiments.

We have provided proofs of all of our Theorems in Appendix A.

We have shared our codes for the simulation and real-world data in the supplementary material. The simulation experiments can be run using the simulation.ipynb file, and is self-contained (needed modules are provided in the zip). Due to the large size of the real-world vector datasets, we are unable to share them, but we have shared the code used to run the experiments.

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

## A    STRUCTURES AND RESULTS IN THE MCPC-BLOCK MODEL

In this section, we provide theoretical support to the observed unbalancedness of centrality measures as well as the balancedness and efficacy of relative centrality as discussed in Section 3 in the random graph model. We first reintroduce the model for ease of following.

**The generative block model**    We are interested in the random graph generated by a 4-block model. In this model, we are given a set of $n$ vertices $V = \{v_1, \ldots, v_n\}$ that has a partition into communities $V_0, V_1$, and each community $V_\ell$ has a further partition into a core $V_{\ell,1}$ and the periphery $V_{\ell,0}$. We focus on the balanced case, where $|V_{\ell,c}| = n/4$ for any $(\ell, c)$ pair. This is associated with a $4 \times 4$ block-probability matrix $\mathbb{P}$, that is indexed with $(i, j) \in \{0, 1\}^2$. Furthermore, $\mathbb{P}$ is row stochastic, i.e, $\sum_{(\ell', c')} \mathbb{P}[(\ell, c), (\ell', c')] = 1$ for all $(i, j)$ pair.

Then, for each vertex pair $v_i \in V_{\ell,c}$ and $v_j \in V_{\ell,c'}$, we add an $v_i \to v_j$ edge with probability $\mathbb{P}[(\ell, c), (\ell', c')] \cdot \frac{k}{|V_{\ell',c'}|}$.

Then, we work under the following setting.

1. $\mathbb{P}[(\ell, c'), (\ell, 1)]$ is $\Omega(1)$ for all $\ell$ and any $c'$. That is, within a community, a constant fraction of edges going out from a periphery vertex ends up in the corresponding core, and a constant fraction of edges originating in a core vertex remains in the core.

2. $k = \omega(\log n)$.

3. We denote the total degree, in-degree, and out-degree of a vertex $v_i$ with $\deg(v_i)$, $\deg_+(v_i)$ and $\deg_-(v)$ respectively and the out-neighbors of $v_i$ are denoted as $N_G(v_i)$.

We denote the resultant graph-generating process as $\mathsf{BM}(\{V_{\ell,c}\}_{(\ell,c) \in \{0,1\}^2}, \mathbb{P}, k)$

Then, we are interested in the case when BM generates an $(\alpha, \beta) - \mathsf{MCPC}$ structure with $\alpha = \Omega(1)$ and $\beta \ll 1$. Recall that this implies that $\mathsf{CC}_G(V_{i,1}) > \mathsf{CC}_G(V_{i,0}) + \alpha$ for $i \in \{0, 1\}$ and the fraction of the inter-community edges originating in the core is $\beta$ fraction of that of the ones starting in the periphery.

**Preliminary bounds**    We are primarily interested in the degree of the vertices generated by such a graph to quantify the unbalancedness in degree centrality as well as the power of our *relative centrality* approach. We define the map $X : [n] \to \{0, 1\}^2$ s.t $X[i] = (\ell, c) \iff v_i \in V_{\ell,c}$. First we note down the Chernoff concentration bound, which will be useful going forward.

**Theorem A.1** (Chernoff Hoeffding bound (Hoeffding, 1963)). *Let $z_1, \ldots, z_n$ be i.i.d random variables that can take values in $\{0, 1\}$, with $\mathbb{E}[z_i] = p$ for $1 \le i \le n$. Then we have*

1. $\Pr\left(\frac{1}{n} \sum_{i=1}^n z_i \ge p + \epsilon\right) \le e^{-D(p+\epsilon \| p)n}$

2. $\Pr\left(\frac{1}{n} \sum_{i=1}^n z_i \le p - \epsilon\right) \le e^{-D(p-\epsilon \| p)n}$

*where $D(x\|y)$ is the KL divergence of $x$ and $y$.*

Next, we obtain some bounds on out-degree and in-degree of vertices.

**Lemma A.2** (The graph is almost-regular w.r.t. out-degree). *Let $G$ be a graph generated from $\mathsf{BM}(\{V_{\ell,c}\}_{(\ell,c) \in \{0,1\}^2}, \mathbb{P}, k)$. Then, with probability $1 - n^{-4}$, the out degree of any vertex $v_i \in V$ is bounded as*

$$k - o(k) \le \deg_-(v_i) \le k + o(k)$$

*Proof.* Let $e_{i,j}$ denote the indicator random variable indicating which is 1 if there is an edge from $v_i$ to $v_j$, and 0 otherwise.

Then, $\deg_-(v_i) = \sum_{j \in [n]} e_{i,j}$. Furthermore, let $\deg_{(-,S)}(v_i)$ denote $\sum_{j : v_j \in S} e_{i,j}$. Then, from a simple counting argument, we have $\mathbb{E}[\deg_{(-,V_{\ell,c})}(v_i)] = \frac{k}{|V_{\ell,c}|} \cdot |V_{\ell,c}| \cdot \mathbb{P}[X[i], (\ell, c)] = \mathbb{P}[X[i], (\ell, c)] \cdot k$.

Next, by Chernoff bound we have

$$\Pr(\deg_{(-,V_{\ell,c})}(v_i) > \mathbb{E}[\deg_{(-,V_{\ell,c})}(v_i)] + 8 \cdot \sqrt{k}\sqrt{\log n})$$

$$\leq e^{-D\left(\mathbb{E}[\deg_{(-,V_{\ell,c})}(v_i)]/|V_{\ell,c}| + 8\cdot\sqrt{k}\sqrt{\log n}/|V_{\ell,c}| \,\middle\|\, \mathbb{E}[\deg_{(-,V_{\ell,c})}(v_i)]/|V_{\ell,c}|\right)\cdot|V_{\ell,c}|}$$

We note that the KL divergence $x, y$ is $D(x||y) = x\ln(x/y) + (1-x)\ln((1-x)/(1-y))$, which is lower bounded by $(x-y)^2/2x$ when $x > y$. Then, the KL divergence can be lower bounded as

$$D\left(\mathbb{E}[\deg_{(-,V_{\ell,c})}(v_i)]/|V_{\ell,c}| + 8\cdot\sqrt{k}\sqrt{\log n}/|V_{\ell,c}| \,\middle\|\, \mathbb{E}[\deg_{(-,V_{\ell,c})}(v_i)]/|V_{\ell,c}|\right)|V_{\ell,c}|$$

$$= D\left(\mathbb{E}[\deg_{(-,V_{\ell,c})}(v_i)]/0.25n + 8\cdot\sqrt{k}\sqrt{\log n}/0.25n \,\middle\|\, \mathbb{E}[\deg_{(-,V_{\ell,c})}(v_i)]/0.25n\right)0.25n$$

$$\geq \frac{64k\log n}{2\mathbb{E}[\deg_{(-,V_{\ell,c})}(v_i)]}$$

$$\geq \frac{64k\log n}{k\mathbb{P}[X[v_i],(\ell,c)]}$$

$$\geq 16\log n \qquad \left[\sum_{\ell,c}\mathbb{P}[X[v_i],(\ell,c)] \leq 1\right]$$

That is,

$$\Pr(\deg_{-}^{V_{\ell,c}}(v_i) > k\mathbb{P}[X[v_i],(\ell,c)] + 8\sqrt{k}\sqrt{\log n}) \leq n^{-16}$$

Applying it for all $(\ell,c)$ and summing up we get

$$\Pr(\deg_{-}(v_i) \geq k\cdot\sum_{(\ell,c)}\mathbb{P}[X[i],(\ell,c)] + 32\sqrt{k}\log n) \leq n^{-15}$$

Here we note that $\sum_{(\ell,c)}\mathbb{P}[X[i],(\ell,c)] = 1$ and that $\sqrt{k}\sqrt{\log n} = o(k)$ as $k = \omega(\log n)$. This completes the upper bound. The lower bound follows similarly.

$\square$

Then, the distribution of the in-degree of the vertices can be obtained as follows.

**Lemma A.3.** Let $G$ be a graph generated from $\mathrm{BM}(\{V_{\ell,c}\}_{(\ell,c)\in\{0,1\}^2}, \mathbb{P}, k)$. Then, for any vertex $v_i \in V$, the expected in-degree of $v_i$ is

$$\mathbb{E}\left[\deg_{+}(v_i)\right] = k\cdot\sum_{(\ell',c')}\mathbb{P}[(\ell',c'),X[i]]$$

*Proof.* Let the vertices of $V$ be denoted as $v_1, \ldots, v_n$. Let $e_{i,j}$ denote the indicator random variable indicating which is 1 if there is an edge from $v_i$ to $v_j$, and 0 otherwise.

First, note that all $e_{i,j}$ are independent. Then, $\Pr(e_{i,j}=1) = \frac{\mathbb{P}[X[i],X[j]]\cdot k}{|V_{X[j]}|}$. Then, summing on the expectation we get, for any $v_i$

$$\mathbb{E}[deg_{+}(v_i)] = \sum_{j\in[n]}\mathbb{E}[e_{j,i}] = \sum_{(\ell',c')\in\{0,1\}^2}|V_{\ell',c'}|\cdot\frac{\mathbb{P}[(\ell',c'),X[j]]\cdot k}{|V_{X[i]}|}$$

$$= \sum_{(\ell',c')}\frac{n}{4}\cdot\frac{\mathbb{P}[(\ell',c'),X[j]]\cdot k}{n/4} = k\cdot\sum_{(\ell',c')}\mathbb{P}[(\ell',c'),X[i]]$$

$\square$

Then, we have the following bound on the indegree of the vertices.

**Lemma A.4.** Let $G$ be a graph generated from $\mathsf{BM}(\{V_{i,j}\}_{(i,j)\in\{0,1\}^2}, \mathbb{P}, k)$. Then, the in-degree of any vertex $v$ is bounded as

$$|\Pr(|\deg_+(v) - \mathbb{E}[\deg_+(v)]| \geq 8 \cdot \sqrt{k}\sqrt{\log n}) \leq n^{-16}$$

*Proof.* We obtain this using Theorem A.1.

For the upper tail, we have

$$\Pr(\deg_+(v) > \mathbb{E}[\deg_+(v)] + 8 \cdot \sqrt{k}\sqrt{\log n}) \leq e^{-D(\mathbb{E}[\deg_+(v)]/n + 8\cdot\sqrt{k}\sqrt{\log n}/n \| \mathbb{E}[\deg_+(v)]/n)n}$$

We note that the KL divergence between Bernoulli random variables $x, y$ $D(x\|y) = x\ln(x/y) + (1 - x)\ln((1-x)/(1-y))$, which is lower bounded by $(x-y)^2/2x$ when $x > y$. Then, $D(\mathbb{E}[\deg_+(v)]/n + 4 \cdot \sqrt{k}\sqrt{\log n} \| \mathbb{E}[\deg_+(v)]/n)n$ can analyzed as

$$D(\mathbb{E}[\deg_+(v)]/n + 8 \cdot \sqrt{k}\sqrt{\log n} \| \mathbb{E}[\deg_+(v)]/n)n$$
$$\geq \frac{64k\log n}{2\mathbb{E}[\deg_+(v)]}$$
$$\geq \frac{64k\log n}{k \cdot \sum_{\ell,c}\mathbb{P}[(\ell,c), X[v]]} \geq 16\log n \qquad \left[\sum_{\ell,c}\mathbb{P}[(\ell,c), X[v] \leq 4\right]$$

Then, substituting, we get an upper bound of $n^{-16}$. $\qquad\square$

This, along with the fact that the sum of entries of any column of $\mathbb{P}$ is $\Omega(1)$, gives the following fact.

**Fact A.5.** Let $G$ be a graph sampled from $\mathsf{BM}(\{V_{\ell,c}\}_{(\ell,c)\in\{0,1\}^2}$ where $k = \omega(\log n)$ and $\mathbb{P}[(\ell,c),(\ell,1)] = \Omega(1)$. Then for all vertices $v_i \in V_{\ell,1}$, we have $\deg_+(v) = \Omega(k)$ with probability $1 - n^{-7}$.

*Proof.* This is straight from the fact that $\mathbb{E}[\deg_+(v_i)] = \Omega(k)$ and the tail deviation is $\sqrt{k}\sqrt{\log n} = o(k)$ with high probability. $\qquad\square$

Then we make a connection between the concentration of any of the core/periphery blocks and degree of each vertices in the core.

**Lemma A.6.** Let $G$ be a graph sampled from $\mathsf{BM}(\{V_{\ell,c}\}_{(\ell,c)\in\{0,1\}^2}, \mathbb{P}, k)$ where $k = \omega(\log n)$. Then, for any vertex $v_i \in V_{\ell,c}$ we have with probability $1 - n^{-4}$,

$$\deg_+(v_i) = k(1 \pm o(1)) + k \cdot \mathsf{CC}_G(V_{\ell,c})(1 \pm o(1)).$$

*Proof.* Let us first recall the definition of concentration. We have $\mathsf{CC}_G(S) = (E(\bar{S}, S) - E(S, \bar{S}))/E(S, V)$.

First, we note that $E(S, V) = k(1 \pm o(1)) \cdot |V_{\ell,1}|$. Also, let $\deg_{(+,S)}(u)$ denote the number of edges connected to $u$ from the set $S$.

Then, we have

$$\mathsf{CC}_G(V_{\ell,c}) = \frac{\sum_{v_i \in V_{\ell,c}} \deg_+(v_i) - \sum_{v_i \in V_{\ell,c}} \deg_{(+,V_{\ell,c})}(v_i) - \sum_{v_i \in \overline{V_{\ell,c}}} deg_{(+,V_{\ell,c})}(v_i)}{k(1 \pm o(1)) \cdot |V_{\ell,c}|}$$
$$= \frac{\sum_{v_i \in V_{\ell,c}} \deg_+(v_i) - \sum_{v_i \in V} \deg_{(+,V_{\ell,c})}(v_i)}{k(1 \pm o(1)) \cdot |V_{\ell,c}|}$$
$$= \frac{\sum_{v_i \in V_{\ell,c}} \deg_+(v_i) - k(1 \pm o(1)) \cdot |V_{\ell,c}|}{k(1 \pm o(1)) \cdot |V_{\ell,c}|}$$

That is,

$$\mathsf{CC}_G(V_{\ell,c}) = \frac{1}{|k(1 \pm o(1)) \cdot V_{\ell,c}|} \cdot \sum_{v_i \in V_{\ell,c}} \deg_+(v) - 1 \qquad (2)$$

Then, we can use the fact that the in-degree of each vertex in $V_{\ell,c}$ is bounded tightly from Lemma A.4. Applying an union bound we get that with probability $1 - n^{-6}$, $|\deg_+(v_i) - \deg_+(v_{i'})| \leq 8\sqrt{k}\sqrt{\log n}$ for any $v_i, v_{i'} \in V_{\ell,c}$.

Then, with the same probability, for any $v_i \in V_{\ell,c}$, we have that $|V_{\ell,c} \cdot \deg_+(v_i) - \sum_{v_{i'} \in V_{\ell,c}} deg_+(v_i')| \leq 32\sqrt{k}\sqrt{\log n}|V_{\ell,c}|$.

Furthermore note that as $k = \omega(\log n)$, we have $\sqrt{k}\sqrt{\log n}/k = o(1)$. This implies with probability $1 - n^{-4}$, for any $v_i \in V_{\ell,c}$,

$$\left| \mathsf{CC}_G(V_{\ell,c}) - \left( \frac{|V_{\ell,c}|}{|k(1 \pm o(1)) \cdot V_{\ell,c}|} \cdot \deg_+(v_i) - 1 \right) \right| \leq 32\sqrt{k}\sqrt{\log n}/k$$

$$\implies \left| \mathsf{CC}_G(V_{\ell,c}) - \left( \frac{1}{k(1 \pm o(1))} \cdot \deg_+(v) - 1 \right) \right| = o(1)$$

$$\implies \frac{1}{k(1 \pm o(1))} \cdot \deg_+(v_i) = 1 + \mathsf{CC}_G(V_{\ell,c}) \pm o(1)$$

$$\implies \deg_+(v_i) = k + k \cdot \mathsf{CC}_G(V_{\ell,c}) \pm o(k)\mathsf{CC}_G(V_{\ell,c}) \pm o(k)$$

$$\implies \deg_+(v_i) = k(1 \pm o(1)) + k \cdot \mathsf{CC}_G(V_{\ell,c})(1 \pm o(1)).$$

This completes the proof. □

### A.1 PROOF OF THEOREM 3.2

Now, we are ready to complete our first proof. First, we know that $\deg(v_i) = \deg_+(v) + \deg_-(v)$, where $\deg_-(v) = k \pm o(k)$. Then Lemma A.6 directly implies that with probability $1 - n^{-4}$, $\deg(v_i) = 2k + k\mathsf{CC}_G(V_{\ell,1}) \pm o(k)$.

Here, Fact A.5 dictates $\deg_+(v) = \Omega(k)$, and simply applying to Equation 1, we get $\mathsf{CC}_G(V_{\ell,1}) = \Omega(1)$. Then, we can write that $\deg(v_i) = k \cdot (2 + \mathsf{CC}_G(V_{\ell,1}) \pm o(\mathsf{CC}_G(V_{\ell,1})))$, which completes the proof.

### A.2 MOST CORE POINTS IN THE 1-STEP N-RANK HAVE HIGH SCORE

We first show that all the core points in our block model with MCPC structure are assigned a high score in the 1-step N-Rank algorithm (Algorithm 1).

Recall that for 1-step N-Rank we first obtain the score $F(v_i)$ for all vertices $v_i \in V$. It is easy to see, that $F(v_i) = \deg_+(v_i)$ for all vertices.

Then in the next step, for each vertex $v_i$, we select $S_{v_i}$ as the set of vertices with a higher $F$ score. Then, we obtain $\hat{F}(v_i) = \frac{F(v_i)}{\text{average}_{v_j \in S_{v_i}} F(v_j)}$.

Then, we are ready to prove that $\hat{F}(v_i)$ is between $1 - o(1)$ and 1 for any core vertex $v_i$, which we quantify in the following Theorem.

**Lemma A.7.** Let $G$ be a graph obtained from the MCPC block model with $k = \omega(\log n)$ resulting in an $(\Omega(1), o(1/k)) - \mathsf{MCPC}$-structure w.r.t. to the core-periphery blocks. Let $\hat{F}(v)$ be the score of the vertices as per Algorithm 1 for $t = 1$. Then, for $1 - o(1)$ fraction of core vertices $v_i \in V_{\ell,1}$ for any $\ell$, we have $1 - o(1) \leq \hat{F}_G(v_i) \leq 1$.

*Proof.* Recall that the graph has $(\Omega(1), o(1/k)) - \mathsf{MCPC}$ structure. That is, $\mathsf{CC}_G(V_{\ell,1}) \geq \alpha + V_{\ell',0}$ for any $(\ell, \ell')$ pair where $\alpha > 0$ is a constant.

Let $CC$ be the minimum concentration among the cores. Then, Lemma A.6 dictates, that with probability $1 - n^{-3}$,

$$\text{If } v_i \in V_{\ell,0} \text{ then } \deg_+(v_i) \leq k + k \cdot (CC - \alpha) + o(k)$$

On the other hand, for core vertices we have

$$\text{If } v_i \in V_{\ell,1} \text{ then } \deg_+(v_i) \geq k + k \cdot CC - o(k)$$

This implies that for any $v_i \in V_{\ell,1}$ and $v_j \in V_{\ell',0}$, $F_G(v_i) > F_G(v_j)$ with high probability.

Thus, in Algorithm 1 when we select $S_{v_i}$ for any $v_i \in V_{\ell,1}$, it does not include any periphery vertices. Then, we can show that the final score of all core vertices will be pretty similar.

Consider any $V_{\ell,1}$. We first note that $\beta = o(1/k)$. Then it is easy to see that $\mathbb{P}(V_{\ell,1}, V_{\ell',1}) = q = o(1/k)$ for any $\ell, \ell'$.

First we count the number of vertices in $V_{\ell,1}$ that has an outgoing edge to $V_{\ell',1}$ for some $\ell' \neq \ell$ (that is inter-core edges originating in $V_{\ell,1}$. This can be upper bounded as $\sum_{v_i \in V_{\ell,1}} \deg_{(+,V_{\ell',1})}(v_i)$. This sum has an expected value of $|V_{\ell,1}| \cdot \frac{|V_{\ell',1}|k \cdot o(1/k)}{|V_{\ell',1}|} = o(|V_{\ell,1}|)$. With high probability this can also be upper bounded by $o(|V_{\ell,1}|) + \sqrt{|V_{\ell,1}|}\sqrt{\log n} = o(|V_{\ell,1}|)$.

Let $S_{\ell,1}$ denote the vertices in $V_{\ell,1}$ that *do not* have any outgoing edge to $V_{\ell',1}$. For any such vertex, $\max\{\hat{F}(v_j)\}_{v_j \in N_G(v_i)} \leq k \cdot \sum_{(\ell',c')} \mathbb{P}[(\ell',c'), X[i]] + o(k)$.

On the other hand, $F_G(v_i) \geq k \cdot \sum_{(\ell',c')} \mathbb{P}[(\ell',c'), X[i]] - o(k)$.

Then we have

$$\begin{aligned}
\hat{F}(v_i) &\geq \frac{k \cdot \sum_{(\ell',c')} \mathbb{P}[(\ell',c'), X[i]] - o(k)}{k \cdot \sum_{(\ell',c')} \mathbb{P}[(\ell',c'), X[i]] + o(k)} \\
&= 1 - \frac{2 \cdot o(k)}{k \cdot \sum_{(\ell',c')} \mathbb{P}[(\ell',c'), X[i]] + o(k)} \\
&\geq 1 - o(1) \qquad\qquad\qquad \left[\text{As we know } k \cdot \sum_{(\ell',c')} \mathbb{P}[(\ell',c'), X[i]] = \Omega(k)\right]
\end{aligned}$$

This completes our proof. $\qquad\qquad\qquad\qquad\qquad\qquad\qquad\qquad\qquad\qquad\qquad\square$

### A.3 N-RANK HAS HIGH CORE PRIORITIZATION

Next, we show that the periphery vertices have a lower score than almost all core vertices, which indicates a high core prioritization.

**Lemma A.8** (Separation between $\hat{F}$ score of cores and peripheries)**.** Let $G$ be a graph sampled from $\mathsf{BM}(\{V_{\ell,c}\}_{(\ell,c)\in\{0,1\}^2}, \mathbb{P}, k)$ where $k = \omega(\log n)$. Then, for any $v_i \in V_{\ell,0}$ (a periphery vertex), $\hat{F}(v_i) < \min_{v_j \in V_{\ell,1}} \hat{F}(v_j)$ with high probability.

*Proof.* Let $v_i \in V_{\ell,0}$ be a periphery vertex s.t. $\mathsf{CC}(V_{\ell,0}) = CC$. Then, its neighbors consist of some core vertices and other periphery vertices.

Let $k_1$ be the number of neighbors of $v_i$ that is a core vertex. As $\mathbb{P}[(\ell,0), (\ell,1)] = \Omega(1)$, we know $k_1 = \Omega(k)$.

Furthermore, if $v_j \in V_{\ell,1}$, $\deg_+(v_j)$ (which is its $F(v_j)$ score) is lower bounded by $k + k \cdot (CC + \alpha) - o(k)$. On the other hand, $\deg_+(v_j) \leq k + k \cdot CC + o(k)$.

Then, $\text{average}_{v_j \in S_{v_i}} F(v_j)$ is lower bounded by

$$\text{average}_{v_j \in S_{v_i}} F(v_j) \geq \frac{k_1 \cdot \min\limits_{v_j \in S_{v_j} \cap V_{\ell,1}} \deg_+(v_j) + (\deg_-(v_i) - k_1) \cdot \deg_+(v_i)}{\deg_-(v_i)}$$

$$\geq \deg_+(v_i) + \frac{k_1}{\deg_{v_i}} \cdot \left( \min\limits_{v_j \in S_{v_j} \cap V_{\ell,1}} \deg_+(v_j) - \deg_+(v_i) \right)$$

$$\geq \deg_+(v_i) + \frac{k_1}{\deg_{v_i}} \cdot (k \cdot \alpha - o(k)) \quad \text{[Using the lower bound on } \deg_+(v_j) : v_j \in V_{\ell,1}$$

$$\text{and upper bound on } \deg_+(v_i)]$$

$$\geq \deg_+(v_i) + C_3 \cdot k \quad \text{[For some constant } C_3]$$

$$\geq \deg_+(v_i) + C_4 \cdot \deg_+(v_i) \quad \text{[For some constant } C_4 \text{ as } \deg_+(v_i) = \mathcal{O}(k)]$$

$$\geq \deg_+(v_i)(1 + C_4)$$

Then, we have $\hat{F}(v_i) \leq \frac{\deg_+(v_i)}{\deg_+(v_i)(1+C_4)} \leq \frac{1}{1+C_4}$ with probability $1 - o(1/n)$. That is, $\hat{F}(v_i)$ is upper bounded by a constant less than 1 with very high probability. On the other hand, Lemma A.7 dictates that $\hat{F}(v_i)$ value of core vertices is upper bounded by $1 - o(1)$. Then there exists a large enough $n_0$ such that for all $n \geq n_0$ there is a separation. $\qquad\square$

## A.4 EXPECTED BEHAVIOR OF THE TOP-RANKED POINTS OF N-RANK

So far, we have shown that while the degree of a core vertex is directly proportional to its concentration, the N-Rank score of core vertices is high $(1 - o(1))$ irrespective of the concentration of the core.

Now we prove Theorem 3.3, which shows that on expectation, roughly the same fraction $\Theta(1/k)$ of points from each core have a score of 1. This is a provable improvement on degree centrality in terms of balancedness.

### A.4.1 FACTS ON THE BEHAVIOR OF THE IN-DEGREE OF THE VERTICES IN AN MCPC BLOCK MODEL GRAPH

First, we make some further observations (and recall some previously made ones) about the in-degree and the out-degree neighbors of the vertices. For simplicity, we define $\hat{\mathbb{P}}([\ell, c], [\ell', c']) := \frac{k}{|V_{\ell',c'}|} \cdot \mathbb{P}([\ell, c], [\ell', c'])$.

1. The in-degree of any vertex can be seen as a sum of $n$ independent Bernoulli random variables (the possibility of edges), with the $\deg_+(v_i)$ having the same distribution for all $v_i \in V_{\ell,c}$ as follows.
   Let $\text{Binom}(n', p')$ refer to the sum of $n'$ i.i.d Bernoulli variables with probability $p'$ of outputting 1. Then,

   $$\deg_+(v_i) \sim \sum_{(\ell',c') \in \{0,1\}^2} \text{Binom}\left( |V_{\ell',c'}|, \hat{\mathbb{P}}[(\ell', c'), (\ell, c)] \right) \qquad (3)$$

2. For any core vertex $v_i \in V_{\ell,1}$, $\deg_+(v_i)$ is greater than all its out-degree neighbors from a periphery (this is straightforward to see from Lemma A.8) with probability $1 - o(1/n)$.

3. For any core-vertex $v_i \in V_{\ell,1}$, probability that it has an out-going edge to $V_{\ell',1}$ is $o(1)$.

4. Without loss of generality, we shall prove the results for the core $V_{0,1}$. The same result applies to any core (of course, the dependencies on $\mathbb{P}$ change based on which core we consider).

Next, we prove an useful relation between the degree of a vertex and degree of a neighbor.

**Weak correlation between in-degree of a vertex and in-degree of their neighbors.**

We note that the in-degrees of $u \in N_G(v) \cap V_{0,1}$ ($N_v$) are slightly different from Equation (3). This is because one edge $v \to u$ is deterministically there. For simplicity let $\tilde{p}_{\ell,c} = \hat{\mathbb{P}}[(\ell,c),(0,1)]$. Then we have

1. $\deg_+(v) \sim X \equiv \underbrace{\mathsf{Binom}(|V_{0,1}|, \tilde{p}_{0,1})}_{X_{0,1}} + \sum_{(\ell,c) \in \{0,1\}^2 \setminus \{0,1\}} \mathsf{Binom}(|V_{\ell,c}|, \tilde{p}_{\ell,c})$      (as before).

2. $\deg_+(u) \sim 1 + Y \equiv 1 + \underbrace{\mathsf{Binom}(|V_{0,1}| - 1, \tilde{p}_{0,1})}_{Y_{0,1}} + \sum_{(\ell,c) \in \{0,1\}^2 \setminus \{0,1\}} \mathsf{Binom}(|V_{\ell,c}|, \tilde{p}_{\ell,c})$

    (slightly modified)

Next, we note down the following useful equality

$$\Pr(\mathsf{Binom}(n', p') = t) = \frac{n' \cdot p'}{t} \cdot \Pr(\mathsf{Binom}(n' - 1, p') = t - 1)$$

$$\implies \Pr(\mathsf{Binom}(n', p') = t) = \frac{\mathbb{E}[\mathsf{Binom}(n', p')]}{t} \cdot \Pr(\mathsf{Binom}(n' - 1, p') = t - 1)$$

Further, note that $\mathbb{E}[X_{0,1}] = k \cdot \mathbb{P}[(0,1),(0,1)]$. Further, as $k = \omega(\log n)$ and $\mathbb{P}[(0,1),(0,1)] = \Omega(1)$, a simple Chernoff bound applied along the lines of Lemma A.4 implies that there exists $\epsilon = o(1)$ such that

$$\Pr\left((1 - \epsilon)\mathbb{E}[X_{0,1}] \leq X_{0,1} \leq (1 + \epsilon)\mathbb{E}[X_{0,1}]\right) \geq 1 - n^{-8} \tag{4}$$

Similarly,

$$\Pr\left((1 - \epsilon)\mathbb{E}[Y_{0,1}] \leq Y_{0,1} \leq (1 + \epsilon)\mathbb{E}[Y_{0,1}]\right) \geq 1 - n^{-8} \tag{5}$$

Let this range, $[(1 - \epsilon)\mathbb{E}[X_{0,1}], (1 + \epsilon)\mathbb{E}[X_{0,1}]]$ be called $R_\epsilon^x$. Similarly we define $R_\epsilon^y$ w.r.t. $Y_{0,1}$. Then,

$$(1 - \epsilon)\Pr(Y = t - 1 | Y_{0,1} \in R_\epsilon^y) \leq \Pr(X = t | X_{0,1} \in R_\epsilon^x) \leq (1 + \epsilon)\Pr(y = t - 1 | Y_{0,1} \in R_\epsilon^y) \tag{6}$$

**A result on bounding functions of pmfs.**

As we shall see, a key term we shall control in our analysis would be the following kinds of sum, specifically $\sum_{i=0}^{n} \Pr(X = t) \cdot \Pr(X \leq t)^k$ and $\sum_{i=0}^{n} \Pr(x = t) \cdot \Pr(X < t)^k$ where $X$ is a discrete distribution with the support $\{0, \ldots, n\}$. In our analysis, $X$ will be the r.v. that describes the in-degree of a vertex.

In this direction, we prove the following useful lemma.

**Lemma A.9.** Let $X$ be any discrete random variable with the support set $\{0, \ldots, n\}$. Then, the following holds for any $k \leq n$.

$$\sum_{x=0}^{n} \Pr(X = x) \cdot \Pr(X < x)^k - 1/n \leq \frac{1}{k+1} \leq \sum_{x=0}^{n} \Pr(X = x) \cdot \Pr(X \leq x)^k + 1/n \tag{7}$$

*Proof.* We know $X$ is a discrete random variable with $n$ possible outcomes. Then, we can extend the probability distribution function of $x$ to a continuous function $\hat{f}$ such that $\hat{f}(\hat{t}) = \Pr(x = t)$ for any $\hat{t} \in [t, t + 1)$. Then $\hat{f}$ can be approximated by a *continuous* function $f$ such that $\int_0^{n+1} |f(x) - \hat{f}(x)| dx \leq \frac{1}{n^c}$ for any constant $c$. For now, let us continue with $c = 4$

This further implies that $\int_0^{n+1} f(x) dx = 1 \pm 1/n^4$. This is because $\int_0^{n+1} \hat{f}(x) dx = 1$ (as $\hat{f}$ a step function representation of a probability distribution function (corresponding to $x$).

Next note that

$\int_0^x \hat{f}(t) dt - 1/n^4 \leq \int_0^x f(t) dt \leq \int_0^x \hat{f}(t) dt + 1/n^4$.

Furthermore, $\Pr(X \leq t) = \int_0^{t+1} \hat{f}(t) dt$ for any $t \in [n]$. This is simple to see as $\Pr(X \leq 0) = \int_0^1 \hat{f}(t) dt$ and so on. Therefore,

$$\Pr(X \leq \lfloor x \rfloor) \leq \int_0^x \hat{f}(t) dt \leq \Pr(X \leq \lceil x \rceil)$$

$$\implies \Pr(X \leq \lfloor x \rfloor) - 1/n^4 \leq \int_0^x f(t) dt \leq \Pr(X \leq \lceil x \rceil) + 1/n^4$$

Then, we define $F(x) := \int_0^x f(t) dt$ as the anti-derivative of $f$. Simply replacing the definition in the equation above gets us

$$\Pr(X \leq \lfloor x \rfloor) - 1/n^2 \leq F(x) \leq \Pr(X \leq \lceil x \rceil) + 1/n^2$$

Also note that $\Pr(X \leq x), \Pr(X < x), f(x),$ and $\hat{f}(x)$ are all *non-decreasing* functions on $x$.

Now, we are ready to prove the bounds. On the right side, we have

$$\sum_{x=0}^{n} \Pr(X = x) \cdot \Pr(X < x)^k$$

$$= \int_0^{n+1} \hat{f}(x) \left( \int_0^{\lfloor x \rfloor} \hat{f}(t) dt \right)^k dx$$

$$\leq \int_0^{n+1} \hat{f}(x) \left( \int_0^x \hat{f}(t) dt \right)^k dx$$

$$\leq \int_0^{n+1} f(x) \left( \int_0^x f(t) dt + 1/n^4 \right)^k dx$$

$$\leq \int_0^{n+1} f(x) \left( \left( \int_0^x f(t) dt + \right)^k + k/n^4 \right) dx$$

$$\leq \int_0^{n+1} f(x) \left( \int_0^x f(t) dt + \right)^k dx + \int_0^{n+1} k/n^4 dx$$

$$\leq \int_0^{n+1} f(x) F(x)^k dx + 1/n^2$$

Next, note that

$$\int_0^{n+1} f(x) F(x)^k dx = \left. \frac{F(x)^{k+1}}{k+1} \right|_0^n = \frac{1}{k+1} \cdot (F[n] - F[0]) \leq \frac{1 + 1/n^4}{k+1} \tag{8}$$

Putting it on the equation above we get

$$\sum_{x=0}^{n} \Pr(X = x) \cdot \Pr(X < x)^k \leq \frac{1}{k+1} + 1/n^2 + 1/n^4 \leq \frac{1}{k+1} + 2/n^2$$

$$\implies \sum_{x=0}^{n} \Pr(X = x) \cdot \Pr(X < x)^k - 2/n^2 \leq \frac{1}{k+1}$$

Next, we prove the other direction. We have

$$\sum_{x=0}^{n} \Pr(X = x) \cdot \Pr(X \le x)^k$$

$$= \int_{0}^{n+1} \hat{f}(x) \left( \int_{0}^{\lceil x \rceil} \hat{f}(t) dt \right)^k dx$$

$$\ge \int_{0}^{n+1} \hat{f}(x) \left( \int_{0}^{x} \hat{f}(t) dt \right)^k dx$$

$$\ge \int_{0}^{n+1} f(x) \left( \int_{0}^{x} f(t) dt - 1/n^4 \right)^k dx$$

$$\ge \int_{0}^{n+1} f(x) \left( \left( \int_{0}^{x} f(t) dt + \right)^k - k/n^4 \right) dx$$

$$\ge \int_{0}^{n+1} f(x) \left( \int_{0}^{x} f(t) dt + \right)^k dx - \int_{0}^{n+1} k/n^4 dx$$

$$\ge \int_{0}^{n+1} f(x) F(x)^k dx - 1/n^2$$

Then applying Equation (8) we get

$$\sum_{x=0}^{n} \Pr(X = x) \cdot \Pr(X < x)^k \ge \frac{1}{k+1} - 2/n^2$$

$$\implies \frac{1}{k+1} \le \sum_{x=0}^{n} \Pr(X = x) \cdot \Pr(X < x)^k + 2/n^2$$

This completes the proof.

$\square$

Next, we shall the aforementioned results to prove an upper bound on the number of core vertices in a core with N-Rank score 1.

### A.4.2 Upper bound on the expected number of core vertices with N-Rank score of 1

Without loss of generality, let $v$ be any core vertex in $V_{0,1}$. To further simplify notation we denote $\tilde{p}_{\ell,c} := \hat{\mathbb{P}}[(\ell, c), (0, 1)]$.

We obtain an upper bound on the probability $\Pr(\hat{F}(v) = 1)$ for any $v \in V_{0,1}$. Let this be defined by the indicator random variable $\mathbf{1}_v$, which is 1 if $v \in S$ and 0 otherwise. Then we have $\mathbb{E}[\#v : v \in V_{0,1}, \hat{F}(v) = 1] = |V_{0,1}| \cdot \mathbb{E}[\mathbf{1}_v]$. We focus on a specific case (that is simpler to tackle).

$$\mathbb{E}[\mathbf{1}_v] = \Pr(\mathbf{1}_v = 1)$$

$$= \Pr\left( \deg_+(v) \ge \max_{u \in N_G(v)} \deg_+(v) \right)$$

$$\le \Pr\left( \deg_+(v) \ge \max_{u \in N_G(v) \cap V_{0,1}} \deg_+(v) \right) \qquad \text{[Only compare with neighbors from the same core]}$$

For the sake of abbreviation we define $N_v := N_G(v) \cap V_{0,1}$.

We shall further condition on the number of neighbors $v$ has that has the same degree as $v$. The reason for this step is more technical. Essentially, we want to use the inequality of Lemma A.9, but we have an upper bound only on sum of terms that look like $\Pr(X = t)(\Pr(X < t))^k$. Here roughly $\Pr(X = t)$ controls $\deg_+(v)$ being $t$, and $(\Pr(X < t))^k$ controls the probability that $k$ of its neighbors have lower degree value. Therefore, we need to consider the case of the neighbors that may have the same degree. In this direction, we define $Z(v) := \{u : u \in N_G(v) \cap V_{0,1}, \deg_+(u) = \deg_+(v)\}$, and get

$$
\begin{aligned}
&\Pr\left(\deg_+(v) \geq \max_{u \in N_G(v) \cap V_{0,1}} \deg_+(u)\right) \\
&= \Pr\left(\deg_+(v) \geq \max_{u \in N_v} \deg_+(u)\right) \\
&= \Pr\left(|Z(v)| \leq \alpha\right) \cdot \Pr\left(\deg_+(v) \geq \max_{u \in N_v} \deg_+(u) \Big| Z(v) \leq \alpha\right) \\
&\qquad\qquad + \\
&\quad\, \Pr\left(|Z(v)| > \alpha\right) \cdot \Pr\left(\deg_+(v) \geq \max_{u \in N_v} \deg_+(u) \Big| Z(v) > \alpha\right) \\
&\leq \Pr\left(\deg_+(v) \geq \max_{u \in N_v} \deg_+(u) \Big| Z(v) \leq \alpha\right) + \Pr\left(|Z(v)| > \alpha\right)
\end{aligned}
\tag{9}
$$

Then, we prove the two following lemmas.

**Lemma A.10.** Let $Z(v) := \{u : u \in N_G(v) \cap V_{0,1}, \deg_+(u) = \deg_+(v)\}$ that is the neighbors of $v$ in $V_{0,1}$ with the same in-degree as $v$. Then, $\Pr(|Z(v)| > 10e \cdot k^{2/3}) = o(1/k)$

*Proof.* First recall that $\deg_+(v)$ is a sum of $n$ independent random Bernoulli variables with $\mathbb{E}[\deg_+(v)] \leq C \cdot k$ for some constant $C$. Then, it is easy to infer that $\max_{i \in \{0,n\}} \Pr(\deg_+(v) = i) \leq \frac{2C}{\sqrt{k}}$. Furthermore, note that $\Pr(\deg_+(v) \notin [\mathbb{E}[\deg_+(v)] - 4\sqrt{k}\sqrt{\log n}, \mathbb{E}[\deg_+(v)] + 4\sqrt{k}\sqrt{\log n}] \leq \frac{1}{n^4}$.

Furthermore with probability $1 - 1/n^4$, $|N_G(v) \cap V_{0,1}| \leq (1 + o(1))k$. Then conditioned on this event, we first bound the probability $\Pr(|Z(v)| = \alpha)$.

We get

$$
\begin{aligned}
\Pr(|Z(v)| = \alpha) &\leq \binom{k}{\alpha} 20 \cdot \sqrt{k} \cdot \sqrt{\log n} \cdot \left(\frac{10}{\sqrt{k}}\right)^\alpha \\
&\leq 20 \cdot \sqrt{k} \cdot \sqrt{\log n} \cdot \left(\frac{e \cdot k}{\alpha}\right)^\alpha \left(\frac{10}{\sqrt{k}}\right)^\alpha \leq 20 \cdot \sqrt{k} \cdot \sqrt{\log n} \cdot \left(\frac{10e \cdot k}{\alpha \sqrt{k}}\right)^\alpha
\end{aligned}
$$

Then, as per the proof statement let $\alpha = 10e \cdot k^{2/3}$. Then we get,

$$
\begin{aligned}
&\Pr(|Z(v)| = 10e \cdot k^{2/3}) \\
&\leq 20 \cdot \sqrt{k} \cdot \sqrt{\log n} \cdot \left(\frac{1}{k^{1/3}}\right)^{k^{2/3}} \\
&\leq 20 \cdot \sqrt{k} \cdot \sqrt{\log n} \cdot \frac{1}{k^2} \cdot \left(\frac{1}{k^{1/3}}\right)^{k^{2/3}-6} \leq \frac{1}{k} \cdot \left(\frac{1}{k^{1/3}}\right)^{k^{2/3}-6}
\end{aligned}
$$

Then, $\Pr(|Z(v)| \geq 10e \cdot k^{2/3}) \leq k \cdot \Pr(|Z(v)| = 10e \cdot k^{2/3}$ (as $\Pr(|Z(v)| = t)$ is a decreasing function on $t$) and we get

$$
\Pr(|Z(v)| \geq 10e \cdot k^{2/3}) \leq \left(\frac{1}{k^{1/3}}\right)^{k^{2/3}-6} \leq \left(\frac{1}{k^{1/3}}\right)^{k^{3/5}} = o(1/k)
$$

$\qquad\qquad\qquad\qquad\qquad\qquad\qquad\qquad\qquad\qquad\qquad\qquad\qquad\qquad\qquad\qquad\qquad\square$

Next we bound the second term, $\Pr\left(\deg_+(v) \geq \max_{u \in N_v} \deg_+(u)\Big|Z(v) \leq Ck^{2/3}\right)$

**Lemma A.11.** Let $v$ be a core vertex. Then

$$\Pr\left(\deg_+(v) \geq \max_{u \in N_v} \deg_+(u)\Big|Z(v) \leq Ck^{2/3}\right) \leq \frac{1 + o(1)}{\mathbb{P}[(0,1),(0,1)]k}$$

*Proof.*

We want to bound $\Pr\left(\deg_+(v) \geq \max_{u \in N_v} \deg_+(u) \mid Z(v) \leq Ck^{2/3}\right)$. Recall that $\mathbb{E}[|N_G(v) \cap V_{0,1}| = \mathbb{P}[(0,1),(0,1)] \cdot k$. Let this value be denoted as $\hat{k}$ for convenience. Then $|N_G(v) \cap V_{0,1}|$ lies within $(1 \pm o(1))\hat{k}$ with probability $1 - n^{-2}$. Then we get

$$\Pr\left(\deg_+(v) \geq \max_{u \in N_v} \deg_+(u)\Big|Z(v) \leq Ck^{2/3}\right)$$

$$\leq \Pr\left(\deg_+(v) \geq \max_{u \in N_v} \deg_+(u)\Big|Z(v) \leq Ck^{2/3}, |N_v| \geq (1 - o(1))\hat{k}\right) + \Pr(|N_v| \leq (1 - o(1))\hat{k}$$

$$\leq \Pr\left(\deg_+(v) \geq \max_{u \in N_v} \deg_+(u)\Big|Z(v) \leq Ck^{2/3}, |N_v| \geq (1 - o(1))\hat{k}\right) + 1/n^2$$

Now, let the event of $Z(v) \leq Ck^{2/3}, |N_v| \geq (1 - o(1)\hat{k}$ be denoted as $E$. Here note now we are working under the condition that there are *at least* $k' = (1 - o(1))\hat{k} - Ck^{2/3}$ neighbors of $v$ in $V_{0,1}$ that do not have the same degree as $v$.

Then, we have

$$\Pr(\deg_+(v) \geq \max_{u \in N_v} |E) \leq \Pr(\deg_+(v) \geq \max_{u \in N_v} |E, x_{0,1} \in R_\epsilon^x) + \Pr(x_{0,1} \notin R_\epsilon)$$

$$\leq \Pr(\deg_+(v) \geq \max_{u \in N_v} |E, x_{0,1} \in R_\epsilon^x) + 1/n^8 \qquad \text{[From Equation 4]}$$

Continuing, we get

$$\Pr(\deg_+(v) \geq \max_{u \in N_v} \deg_+(u)\Big|E, x_{0,1} \in R_\epsilon^x)$$

$$\leq \sum_{t=1}^n \Pr(X = t | X_{0,1} \in R_\epsilon^x) \cdot \Pr(Y < t - 1)^{k'}$$

$$\leq \sum_{t=1}^n (1 + \epsilon) \Pr(Y = t - 1 | Y_{0,1} \in R_\epsilon^x) \cdot \Pr(Y < t - 1)^{k'} \qquad \text{[From Equation 6]}$$

$$\leq \frac{(1 + \epsilon)}{\Pr(Y_{0,1} \in R_\epsilon^y)} \sum_{t=0}^{n-1} \Pr(Y = t) \cdot (\Pr(y < t))^{k'} \qquad [\Pr(A|B) \leq \Pr(A)/\Pr(B)]$$

$$\leq (1 + 1.1\epsilon) \sum_{t=0}^{n-1} \Pr(Y = t) \cdot (\Pr(y < t))^{k'} \qquad \text{[Substitute } \Pr(Y_{0,1} \in R_\epsilon^y) \text{ from Equation (5)]}$$

Finally, from Lemma A.9 we have $\sum_{t=0}^{n-1} \Pr(Y = t) \cdot (\Pr(y < t))^{k'} \leq \frac{1}{k'+1} + 1/n^2$.

Combining the results we get,

$$\Pr\left(\deg_+(v) \geq \max_{u \in N_v} deg_+(u)\Big||Z(v)| \leq Ck^{2/3}\right)$$

$$\leq \frac{1 + 1.1\epsilon}{\mathbb{P}[(0,1),(0,1)](k - Ck^{2/3})} + 2/n^2$$

$$\leq \frac{1 + o(1)}{\mathbb{P}[(0,1),(0,1)]k}$$

$\square$

Then, continuing from Equation (9) we get

$$\Pr(\deg_+(v) \geq \max_{u \in N_G(v)} \deg_+(u)) \leq \Pr\left(\deg_+(v) \geq \max_{u \in N_v} \deg_+(u) \Big| Z(v) \leq \alpha\right) + \Pr\left(|Z(v)| > \alpha\right)$$

$$\leq \frac{1 + o(1)}{\mathbb{P}[(0,1),(0,1)]k} + o(1/k)$$

$$\leq \frac{1 + o(1)}{\mathbb{P}[(0,1),(0,1)]k}$$

$$\implies \mathbb{E}[|V_{0,1} \cap \{v_i : \hat{F}(v_i) = 1\}|] \leq \frac{1 + o(1)|V_{0,1}|}{\mathbb{P}[(0,1),(0,1)]k}$$

$$(10)$$

Note that as the upper bound holds for any core $V_{\ell,1}$. Next we establish a lower bound, which we find slightly easier to get (and is still within a $(1 - o(1))$ factor of the upper bound).

### A.4.3 LOWER BOUND ON THE EXPECTED NUMBER OF CORE VERTICES WITH N-RANK SCORE OF 1

Next, we want to obtain a lower bound on the expected number of core vertex with $\hat{F}(v) = 1$ *irrespective of the core concentration*. Here, we re-emphasize that our upper bound was also derived in the same manner.

As before, we are interested in

$$\Pr(\mathbf{1}_v = 1) = \Pr\left(\deg_+(v) \geq \max_{u \in N_G(v)} \deg_+(v)\right)$$

Now, we consider two events

1)
$$E_1 := \deg_+(v) \geq \max_{N_G(v) \cap (V_{0,0} \cup V_{1,0})} \deg_+(u)$$

This is the event that $v$ has a higher in-degree than all out-neighbor that is a periphery vertex. Note that this happens with probability $(1 - \mathcal{O}(1/n))$ from Lemma A.8.

2)
$$E_2 := |N_G(v) \cap V_{1,1}| = 0$$

This implies that $v$ has no outgoing edge to another core. Note that $E_2$ happens with probability $1 - o(1)$ as $\beta = o(1/k)$.

Then we have

$$\Pr(\mathbf{1}_v = 1) \geq \Pr(E_1) \Pr(E_2) \Pr(\mathbf{1}_v = 1 \mid E_1, E_2)$$
$$\geq (1 - o(1)) \cdot (1 - \mathcal{O}(1/n)) \cdot \Pr(\mathbf{1}_v = 1 \mid E_1, E_2) \geq (1 - o(1)) \cdot \Pr(\mathbf{1}_v = 1 \mid E_1, E_2)$$

Then, we focus on the conditional probability of the vertex having the highest degree, which is as follows

$$\Pr(\mathbf{1}_v = 1 \mid E_1, E_2) = \Pr\left(\deg_+(v) \geq \max_{u \in N_G(v) \cap V_{0,1}} \deg_+(u)\right)$$

Let $\mathbf{1}_v \mid E_1, E_2 := \hat{\mathbf{1}}_v$. We then focus on this event for a fixed value of $|N_G(v) \cap V_{0,1}|$, i.e., $\hat{\mathbf{1}}_v | |N_G(v) \cap V_{0,1}| = k'$ We can write it down as

$$\mathbb{E}[\hat{\mathbf{1}}_v | |N_G(v) \cap V_{0,1}| = k'] = \sum_{t=0}^{n} \Pr(X = t) \cdot \Pr((Y \leq t - 1))^{k'}$$

As we have a lower bound in Lemma A.9 of this form, we do not need to consider the case of neighbors having the same in degree separately, as we did for the upper bound.

Furthermore, for the upper bound, we switched $X$ to $Y$ through a simple conditional probability upper bound of $\Pr(A|B) \le \Pr(A)/\Pr(B)$. Unfortunately, there is no such trivial lower bound that is usable here. Therefore we need a more precise analysis.

Recall that $X$ is a sum of $2\ell$ many binomial distribution (each corresponding to one of the blocks). In the previous section we used $X_{0,1}$ to obtain an upper bound. Let each of these binomial r.vs be denoted as $X_{\ell,c}$.

We first define the ordered set $T = (t_{0,0}, t_{0,1}, t_{1,0}, t_{1,1})$ and let $T_{\mathsf{sum}} := \sum_{i \in T} i$ denote the sum of the elements in $T$.

Then we have,

$$
\begin{aligned}
\Pr(x = t) &= \sum_{T : T_{\mathsf{sum}} = t} \prod \Pr(X_{\ell,c} = t_{\ell,c}) \\
&= \underbrace{\sum_{\substack{T \,:\, t_{0,1} \in R_\epsilon^x \\ T_{\mathsf{sum}} = t}} \prod \Pr(X_{\ell,c} = t_{\ell,c})}_{\Sigma_1} + \underbrace{\sum_{\substack{T \,:\, t_{0,1} \notin R_\epsilon^X \\ T_{\mathsf{sum}} = t}} \prod \Pr(X_{\ell,c} = t_{\ell,c})}_{\Sigma_2}
\end{aligned}
$$

First we show that $\Sigma_2$ is very small, which is easy to show.

$$
\begin{aligned}
\Sigma_2 &= \sum_{\substack{T \,:\, t_{0,1} \notin R_\epsilon^x \\ T_{\mathsf{sum}} = t}} \prod \Pr(X_{\ell,c} = t_{\ell,c}) \\
&\le \sum_{T : t_{0,1} \notin R_\epsilon^x, T_{\mathsf{sum}} = t} \Pr(X_{0,1} = t_{0,1}) \\
&\le \sum_{T : t_{0,1} \notin R_\epsilon^x, T_{\mathsf{sum}} = t} \frac{1}{n^8} \qquad \text{[Applying Chernoff as in Equation (5)]} \\
&\le \frac{1}{n^4} \qquad\qquad\qquad \left[\begin{array}{l}\text{there are only } 4 \text{ elements} \\ \text{each has } t \le n \text{ possible choices}\end{array}\right]
\end{aligned}
$$

Next, we look at $\Sigma_1$. Our goal is to switch $x$ to $y$ as before (with more care as now we need a lower bound).

$$
\begin{aligned}
\Sigma_1 &= \sum_{\substack{T \,:\, t_{0,1} \in R_\epsilon^x \\ T_{\mathsf{sum}} = t}} \prod \Pr(X_{\ell,c} = t_{\ell,c}) \\
&= \sum_{\substack{T \,:\, t_{0,1} \in R_\epsilon^x \\ T_{\mathsf{sum}} = t}} \Pr(X_{0,1} = t_{0,1}) \cdot \Pr(X_{0,0} = t_{0,0}) \cdot \Pr(X_{1,1} = t_{1,1}) \cdot \Pr(X_{1,0} = t_{1,0}) \\
&= \sum_{\substack{T \,:\, t_{0,1} \in R_\epsilon^x \\ T_{\mathsf{sum}} = t}} \Pr(X_{0,1} = t_{0,1}) \cdot \Pr(Y_{0,0} = t_{0,0}) \cdot \Pr(Y_{1,1} = t_{1,1}) \cdot \Pr(Y_{1,0} = t_{1,0})
\end{aligned}
$$

$$
\text{[As apart from } y_{0,1}, \text{ other } X_{\ell,c} \text{ and } Y_{\ell,c} \text{ are identical ]}
$$

$$
\implies (1 - \epsilon)\Pr(Y = t | y_{0,1} \in R_\epsilon^y) \le \Sigma_1 \le (1 + \epsilon)\Pr(Y = t | y_{0,1} \in R_\epsilon^y)
$$

The last statement comes from the fact that in range of $t_{0,1} - 1 \in R_\epsilon^y$ $(1 - \epsilon)\Pr(Y_{0,1} = t_{0,1} - 1) \le \Pr(X_{0,1} = t_{0,1}) \le (1 + \epsilon)\Pr(Y_{0,1} = t_{0,1} - 1)$.

Furthermore, using the calculation of $\Sigma_2$ directly implies that $\Pr(Y = t | y_{0,1} \in R_\epsilon^y) \ge \Pr(y = t | Y_{0,1}) - 1/n^3$.

Combining these calculations we get:

$$\Pr(X = t) \geq (1 - \epsilon) \Pr(Y = t - 1) - \frac{2}{n^3} \tag{11}$$

Then we have

$$\mathbb{E}[\hat{\mathbf{1}}_v || N_G(v) \cap V_{0,1} = k']$$

$$\geq \sum_{t=1}^{n} \Pr(X = t) \cdot (\Pr(Y \leq t - 1))^{k'}$$

$$\geq \sum_{t=1}^{n} \left( (1 - \epsilon) \Pr(Y = t - 1) - 2/n^3 \right) (\Pr(Y \leq t - 1))^{k'}$$

$$\geq \sum_{t=0}^{n-1} (1 - \epsilon) \Pr(Y = t)(\Pr(Y \leq t))^{k'} - 2/n^2$$

$$\geq (1 - \epsilon) \sum_{t=0}^{n-1} \Pr(Y = t)(\Pr(Y \leq t))^{k'} - 2/n^2$$

$$\geq (1 - \epsilon)\frac{1 - o(1)}{k'} - 2/n^2 \qquad \text{[Applying the lower bound from Lemma A.9]}$$

$$\geq \frac{1 - o(1)}{k'} \qquad \text{[As } \epsilon = o(1)\text{]}$$

Then, compiling all the results we have

$$\mathbb{E}[\mathbf{1}_v] \geq (1 - o(1))(1 - \mathcal{O}(1/n)) \underset{|N_G(v) \cap V_{0,1}|}{\mathbb{E}} \left[ \hat{\mathbf{1}}_v || N_G(v) \cap V_{0,1}| = k' \right]$$

$$\geq (1 - o(1)) \underset{|N_G(v) \cap V_{0,1}|}{\mathbb{E}} \left[ \cdot\frac{(1 - o(1))}{k'} - 2/n^2 \right]$$

$$\geq (1 - o(1)) \underset{|N_G(v) \cap V_{0,1}|}{\mathbb{E}} \left[ \frac{(1 - o(1))}{k' + 1} - 2/n^2 \right]$$

$$\geq \frac{(1 - o(1))}{\mathbb{P}[(0, 1), (0, 1)]k(1 + o(1))}$$

$$\geq \frac{(1 - o(1))}{\mathbb{P}[(0, 1), (0, 1)]k}$$

As we had chosen the core to be $V_{0,1}$ without loss of generality, the result holds for any core. We can thus conclude

$$\mathbb{E}\left[ |V_{\ell,1} \cap \{v_i : \hat{F}(v_i) = 1\}| \right] \geq \frac{(1 - o(1))|V_{\ell,1}|}{\mathbb{P}[(\ell, 1), (\ell, 1)] \cdot k} \tag{12}$$

### A.4.4  PROOF OF THEOREM 3.3

Then, we can summarize the proof directly from the upper bound obtained in Appendix A.4.2 (concluded in Equation (10)) and the lower bound obtained in the Appendix A.4.3 (concluded in Equation (12)).

# B  META ALGORITHM

---

**Algorithm 2** A meta generalization:
Meta-Relative-Rank $(t, y, z)$

---

**Input:** Graph $G(V, E)$ and meta-parameters $t, y, z$.
Obtain $F^{(t)}$ as follows.
$\mathbf{s} \leftarrow \mathbf{1}_n$        {#Vector of all ones}
**for** i in 1:t **do**
    $\mathbf{s} \leftarrow A^T \mathbf{s}$        {# Obtaining initial centrality score}
**end for**
$F^{(t)}(v_i) \leftarrow \mathbf{s}[i]$

{1. Obtain a $y$-hop NeighborRank}

**for** $v \in V$ **do**
    $S_v^{(y)} \leftarrow \{u : u \in N_{G,y}(v), F^{(t)}(u) > F^{(t)}(v)\}$

    $\hat{F}_G^{(t)}(v) \leftarrow \underset{u \in S_v^{(y)}}{\mathsf{average}}[F^{(t)}(u)]/F^{(t)}(v)$

**end for**

{2. Recurse the process $z$ times}

$H_G \leftarrow \hat{F}^{(t)}, counter \leftarrow 0$
**while** $counter < z$ **do**
    **for** $v \in V$ **do**
       $S_v^{(y)} \leftarrow \{u : u \in N_{G,y}(v), H(u) > H(v)\}$

       $\hat{H}(v) \leftarrow \underset{u \in S_v^{(y)}}{\mathsf{average}}[H(u)]/H(v)$

    **end for**
    $H \leftarrow \hat{H}, counter \leftarrow counter + 1.$
**end while**
**return** $\hat{H}$

---

We design the algorithm meta-algorithm by extending the idea of N-Rank (Algorithm 1) in two ways (as we briefly discussed in Section 3.

1) There may be periphery vertices in the graph that have a high $F_G$ value compared to its 1-hop neighborhood. To mitigate this issue, we can look at some $y$-hop neighborhood $N_{G,y}(v)$ of $v$ when selecting the reference set.

As we look at a larger set of vertices for comparison, this method is less likely to report local maxima as core nodes and thus cause *increased core prioritization*. On the other hand, if we look at a vertex from a sparse core, then a large-hop neighborhood may contain more vertices from other cores, and using them in the reference set reduces their final core. This leads to potentially a *lower balancedness*.

2) We have observed that our N-Rank approach increases the balancedness in the initial centrality measure $F_G$. In this direction, we can recursively apply this process by first calculating the $y$-hop N-Rank value and then feeding it back to the algorithm as the initial centrality measure to further increase balancedness. We can apply this recursive process any $z \geq 1$ many times.

The idea is that if the $y$-hop N-Rank has higher balancedness than the initial centrality measure, recursively applying the process should result in *increased balancedness* up to a point. On the other hand, such a method can also amplify any loss of core prioritization due to $y$-hop N-Rank, and thus lead to potentially a *lower core prioritization*.

### B.1 THE CHOICE OF T FOR THE INITIAL CENTRALITY MEASURE

In the first step of our algorithm, we obtain an initial centrality measure $F^{(t)}$ by calculating the total probability of random walks starting at different vertices and reaching a particular vertex. When $t = 1$, this converges to the in-degree of a vertex (up to some multiplicative factor). When $t$ is larger, this can be thought of as a truncated variant of PageRank without any damping.

In our experiments, we set $t = 1$ for the graphs generated by the MCPC block model and $t = \log |V|$ for both concentric GMM as well as real-world experiments. This is based on the intuition that if $t$ is large, then it can help discard local minima. In real-world data, there may be some periphery vertices that do not have any outgoing edges to a core and, as such, can have the highest $F^{(t)}$ value among its outgoing neighbors, thus obtaining a score of 1. A larger $t$ can solve this issue.

On the other hand, if $t$ is too large, it can further increase the $F^{(t)}$ value of a core with higher concentration (as more random walks will be trapped there) and can reduce the balancing effect of the subsequent steps.

We further note that while we choose $t = \log |V|$, the final outcome is not very sensitive to the choice of $t$, and for most graphs, the results for $3 \leq t \leq \log |V|$ were quite similar. Finally, while we do not need to do a hyperparameter tuning for our real-world experiments (fixing $t = \log |V|$ for all experiments), it is an important direction to consider whether a data-dependent value of $t$ can be obtained.

In general, observing the impact of using different initial centrality measures on the overall performance is an important direction.

## C A MORE COMPLEX SIMULATION MODEL: THE CONCENTRIC GAUSSIAN MIXTURE MODEL

Mixture models are one of the fundamental statistical models to study the inference of communities in data. Here, each underlying community has a center, and then points for that community are generated with respect to distributions with mean as the respective center. When the distributions are Gaussian, the model is called the Gaussian mixture model (GMM) (Reynolds et al., 2009), which is widely studied in the clustering literature (Sanjeev & Kannan, 2001; Kumar & Kannan, 2010; Lu & Zhou, 2016; Löffler et al., 2021). In this section, we extend the GMM in a natural way to incorporate an MCPC structure.

**Definition C.1** (Concentric GMM with two communities). There are two centers $\mathbf{c}_\ell \in \mathbb{R}^d, \ell \in \{0, 1\}$. *Each center* is associated with *two* $d$-dimensional isotropic Gaussian distributions with the center as its mean and variances $\sigma_{\ell,1}$ and $\sigma_{\ell,0}$ respectively, with $\sigma_{\ell,1} \geq 1.1\sigma_{\ell,0}$. We denote the distributions as $\mathcal{D}_1^{(\ell)}$ and $\mathcal{D}_0^{(\ell)}$ respectively. We sample $n_{\ell,j}$ points from each distribution $\mathcal{D}_j^{(\ell)}$, and collectively note them as $V_{\ell,j}$. Then, the two underlying communities are $V_\ell = V_{\ell,1} \cup V_{\ell,0}$.

For simulation, we set $d = 20$, $|V_{i,j}| = 2000$ for all $i, j$, and the variances of the distributions are set as $\begin{bmatrix} \sigma_{0,1} & \sigma_{0,0} & \sigma_{1,1} & \sigma_{1,0} \\ 0.1 & 0.3 & \gamma \cdot 0.1 & \gamma \cdot 0.3 \end{bmatrix}$ where $1 \leq \gamma < 3$. Furthermore, we choose the centers as two so that there is an overlap between the two communities. Then, we make the following observations.

i) The 20-NN embedding has an $(\alpha, \beta) - $ MCPC periphery structure, with the points sampled from the lower variance distribution $\sigma_{0,0}$ and $\sigma_{1,1}$ forming the cores and the rest being peripheries.

ii) When $\gamma = 1$, the two communities are symmetrical and we have a $(2, 0.3) - $ MCPC structure with both cores having similar concentration. Then all core-ranking methods have high balancedness, as noted in Figure 7(b), and when we select the top points from the ranking, the induced subgraph has a higher ICEF, as observed in Figure 7(a). From hereon, we set $t := \log n$ step in the first step for all our algorithms to avoid local maxima when obtaining the initial centrality scores.

iii) **Cores with different concentration.** Next, we observe that if we set $\gamma = 1.5$, then one core has a higher variance than the other, and the corresponding vertices in the graph have a lower concentration than the other core. Then, traditional centrality measures indeed have a lower balancedness, and our methods perform relatively better, with RN-Rank having the highest balancedness value, as noted

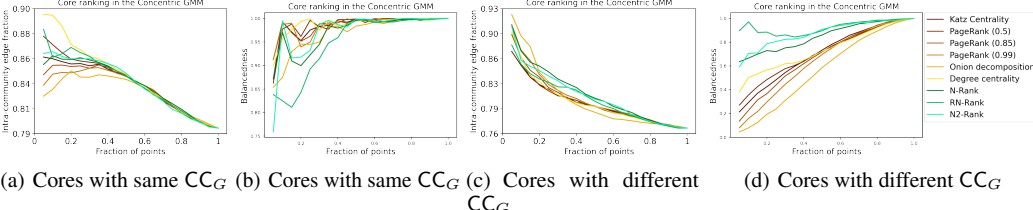

(a) Cores with same $\mathsf{CC}_G$ (b) Cores with same $\mathsf{CC}_G$ (c) Cores with different $\mathsf{CC}_G$ (d) Cores with different $\mathsf{CC}_G$

Figure 7: Improvement in ICEF, and balancedness of of top points for different ranking algorithms in concentric GMM

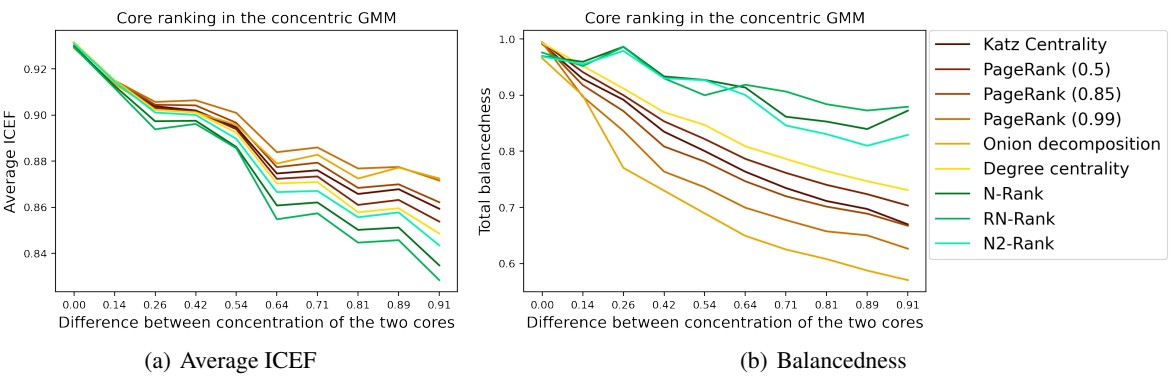

(a) Average ICEF                    (b) Balancedness

Figure 8: Average ICEF and total balancedness of ranking by centrality on the 20-NN embedding of data generated with the concentric GMM for different instantiations of the model

in Figure 7(d). Furthermore, we note that all methods provide a similar increase in ICEF upon core selection in Figure 7(c), with N2-Rank being slightly better than our other methods, the same as in the random graph model. We provide a more detailed simulation result for different values of $\gamma$ in Appendix C to further support the balancedness of relative centrality.

Next, we do a larger-scale simulation to concretize this observation and further look at the ICEF-improvement vs. balancedness tradeoff of our methods. We use the same setup as in Section C, but run our methods on graphs generated with several different parameters.

We set $d = 20$, size of each core or periphery $V_{\ell,c} = 2000$, and the two centers being $\mathbf{c_1} = \{0\}^d$ and $\mathbf{c_2} = 0.3 \cdot \{1\}^d$. Then, we set the following variances of the distributions, parameterized by $\gamma$.

$\sigma_{0,1} = 0.1, \sigma_{1,1} = 0.3$ and $\sigma_{1,c} = \gamma \cdot \sigma 0, c$. Then, we generate $V_{\ell,c}$ many points with a $d$-dimensional isotropic Gaussian with $\mathbf{c_\ell}$ as the center and $\sigma_{\ell,c}$ as variance, with the cores being generated with lower variance distribution than their corresponding periphery.

Then, when $\gamma = 1$, both the cores (and the peripheries) are generated from distributions of the same variance 0.1 (0.3 for the peripheries). As a result, the cores have similar concentrations, and the graph satisfies a $(0.2, 0.75)$-MCPC structure. Then, the average ICEF of the induced subgraph, as well as the total balancedness of the ranking of all methods, are very similar.

Next, we increase $\gamma$ slowly, resulting in the variance of the second community being higher both for the core and the periphery. The results are captured in Figure 8. While the average ICEF of all the methods is pretty similar (within $4\%$), the traditional centrality measures have significantly higher balancedness, becoming as high as $20\%$ for $\gamma = 2$.

Furthermore, among our methods, RN-Rank seems to have the highest balancedness, whereas N2-Rank seems to have the highest ICEF. This tradeoff indicates the possibility of more interesting algorithms via the relative centrality framework going forward.

| Name | Abbreviation | # of points | # of communities | Source |
|------|-------------|-------------|------------------|--------|
| Baron_Human | BM | 8569 | 14 | (Abdelaal et al., 2019) |
| Baron_Mouse | BH | 1886 | 13 | (Abdelaal et al., 2019) |
| Muraro | Mu | 2122 | 9 | (Abdelaal et al., 2019) |
| Segerstolpe | Se | 2133 | 13 | (Abdelaal et al., 2019) |
| Xin | Xi | 1449 | 4 | (Abdelaal et al., 2019) |
| Zhengmix8eq | Zh | 3994 | 8 | (Duò et al., 2018) |
| T-cell dataset | Tcell | 5759 | 10 | (Savas et al., 2018) |
| ALM | ALM | 10068 | 136 | (Smith et al., 2019) |
| AMB | AMB | 12382 | 110 | (Abdelaal et al., 2019) |
| TM | TM | 54865 | 55 | (Abdelaal et al., 2019) |
| VISP | VISP | 15413 | 135 | (Smith et al., 2019) |

Table 3: Details of the scRNA datasets we use

## D  SINGLE-CELL DATA

**Datasets**  First, we provide a detailed description of the datasets we use in Table 3. Note that each dataset has annotated labels that we use to verify the performance of our algorithm. Next, we describe the experimental setup. For each dataset, we first log-normalize it and then apply PCA dimensionality reduction of dimension 50, which is a standard pipeline in the single-cell analysis literature (Duò et al., 2018). Then, we obtain 20-NN embedding of each of the datasets and obtain a ranking of the vertices via both the baseline and traditional centrality measures as well as our relative centrality approach. Then, we select some $c$-fraction of the vertices from the top of the ranking and calculate the intra-community edge fraction of the induced subgraph by these vertices and also the preservation ratio, using the annotated labels.

**ICEF and preservation ratio of all datasets**  We put the change in the intra-community edge fraction as well as the preservation ratio by changing the fraction of points we select in the ranking for all of the 11 single-cell datasets that we consider in this paper in Appendix F. As can be observed from the figures, our methods have comparable improvement in intra-community edge fraction to the baselines. However, our methods generally have a superior preservation ratio. We make the following three observations.

i) Our RN-Rank method provides the best preservation ratio among all the methods, and it has slightly lower ICEF improvement. Here, note that for different values of $c$, the baseline methods, in fact, completely fail to include many more underlying communities than our relative centrality based methods.

ii) Among our methods, N2-Rank provides the highest improvement in ICEF and has a lower preservation ratio than our other methods. In this direction, a better understanding of the preservation ratio-ICEF improvement tradeoffs of our framework is a very interesting future direction.

iii) Finally, for the Zhengmix8eq dataset, the traditional centrality measures do not provide any improvement in ICEF via subset selection, as can be observed in Figure 15. This further points to weaknesses in the traditional centrality measures and the power of our relative centrality framework.

**Balancedness of ranking**  We note that the balancedness values for the datasets Xin, Zheng, Tcell, and ALM are moderate, and we observe the same patterns as with the preservation ratio, with RN-Rank obtaining the best results. The other datasets show that the balancedness AUC values are very small, less than $0.1$. This indicates that at least one cluster is lost for these datasets when we filter out points. We attribute this to these datasets having several very small clusters. Improving our algorithms to have non-negligible (worst case) balancedness in such datasets is an important future direction.

**Purity improvement upon core-ranking based point selection**  Here, we put the ICEF, purity, and preservation ratio of the top $20\%$ points for different CR algorithms for all the datasets.

| Datasets | BH | MU | Se | Xi | Zh | Tcell | ALM | AMB | TM | VISP |
|---|---|---|---|---|---|---|---|---|---|---|
| # of points | 1886 | 2122 | 2133 | 1449 | 3994 | 5759 | 10068 | 12832 | 54865 | 15413 |
| Metrics | PR ICEF Purity | PR ICEF Purity | PR ICEF Purity | PR ICEF Purity | PR ICEF Purity | PR ICEF Purity | PR ICEF Purity | PR ICEF Purity | PR ICEF Purity | PR ICEF Purity |
| Original values | 1.00 0.94 0.93 | 1.00 0.94 0.95 | 1.00 0.92 0.89 | 1.00 0.97 0.99 | 1.00 0.78 0.8 | 1.00 0.72 0.72 | 1.00 0.68 0.44 | 1.0. 0.74 0.46 | 1.00 0.94 0.86 | 1.00 0.69 0.48 |
| Katz | 0.52 1.00 1.00 | 0.77 1.00 1.00 | 0.52 0.99 0.99 | 0.63 1.00 1.00 | 0.92 0.79 0.82 | 0.71 0.80 0.82 | 0.50 0.81 0.69 | 0.61 0.85 0.69 | 0.83 0.99 0.95 | 0.57 0.81 0.64 |
| PageRank (0.5) | 0.52 1.00 1.00 | 0.77 1.00 0.99 | 0.52 0.99 0.98 | 0.67 1.00 1.00 | 0.94 0.79 0.76 | 0.76 0.80 0.81 | 0.55 0.80 0.67 | 0.65 0.84 0.69 | 0.85 0.98 0.95 | 0.60 0.80 0.62 |
| PageRank (0.85) | 0.52 1.00 1.00 | 0.76 1.00 1.00 | 0.53 0.99 0.99 | 0.60 1.00 1.00 | 0.89 0.79 0.80 | 0.70 0.79 0.82 | 0.47 0.82 0.69 | 0.61 0.85 0.70 | 0.81 0.99 0.96 | 0.56 0.81 0.64 |
| PageRank (0.99) | 0.52 1.00 1.00 | 0.75 1.00 1.00 | 0.53 0.99 0.99 | 0.55 1.00 1.00 | 0.84 0.79 0.79 | 0.64 0.78 0.83 | 0.46 0.82 0.70 | 0.58 0.85 0.73 | 0.79 0.99 0.97 | 0.54 0.81 0.65 |
| Onion | 0.31 1.00 1.00 | 0.58 0.99 0.99 | 0.40 0.97 0.92 | 0.46 1.00 1.00 | 0.89 0.79 0.77 | 0.57 0.74 0.77 | 0.37 0.81 0.64 | 0.40 0.86 0.74 | 0.61 0.98 0.94 | 0.45 0.80 0.62 |
| Degree | 0.60 1.00 0.99 | 0.77 0.99 0.99 | 0.53 0.99 0.98 | 0.76 1.00 0.99 | 0.97 0.78 0.78 | 0.80 0.80 0.82 | 0.57 0.79 0.61 | 0.69 0.83 0.66 | 0.88 0.98 0.94 | 0.63 0.79 0.60 |
| RN-Rank | 0.71 0.98 0.97 | 0.82 0.99 0.98 | 0.57 0.98 0.98 | 0.79 0.99 1.00 | 0.89 0.86 0.86 | 0.89 0.80 0.80 | 0.65 0.78 0.61 | 0.73 0.81 0.58 | 0.90 0.98 0.94 | 0.67 0.78 0.59 |
| N2-Rank | 0.52 1.00 1.00 | 0.75 1.00 0.99 | 0.51 0.99 0.99 | 0.74 1.00 1.00 | 0.86 0.87 0.89 | 0.89 0.82 0.85 | 0.57 0.78 0.65 | 0.69 0.82 0.65 | 0.88 0.98 0.93 | 0.62 0.79 0.59 |

Table 4: Preservation ratio, intra-community edge fraction, and purity score of Louvain of top one-third ranked points

**NMI improvement upon core-ranking based point selection**    Then, in Table 5, we observe the improvement in the NMI outcome of Louvain when applied on the top $20\%$ of the ranked points by the different methods, along with the preservation ratio of the selected subset. As with the purity, all core ranking methods give subsets that have similar improvements in the NMI.

| Datasets | BM | BH | MU | Se | Xi | Zh | Tcell | ALM | AMB | TM | VISP |
|---|---|---|---|---|---|---|---|---|---|---|---|
| # of points | 8569 | 1886 | 2122 | 2133 | 1449 | 3994 | 5759 | 10068 | 12832 | 54865 | 15413 |
| Metrics | PR NMI | PR NMI | PR NMI | PR NMI | PR NMI | PR NMI | PR NMI | PR NMI | PR NMI | PR NMI | PR NMI |
| Original values | 1.00 0.75 | 1.00 0.70 | 1.00 0.74 | 1.00 0.67 | 1.00 0.60 | 1.00 0.72 | 1.00 0.46 | 1.0 0.74 | 1.00 0.74 | 1.00 0.82 | 1.00 0.69 |
| Katz | 0.80 0.77 | 0.48 0.75 | 0.73 0.78 | 0.50 0.72 | 0.61 0.58 | 0.90 0.80 | 0.60 0.61 | 0.43 0.86 | 0.51 0.87 | 0.74 0.82 | 0.49 0.85 |
| PageRank (0.5) | 0.80 0.76 | 0.51 0.77 | 0.75 0.78 | 0.50 0.70 | 0.64 0.57 | 0.93 0.77 | 0.69 0.63 | 0.47 0.85 | 0.57 0.86 | 0.76 0.82 | 0.53 0.84 |
| PageRank (0.85) | 0.79 0.77 | 0.49 0.75 | 0.74 0.79 | 0.51 0.72 | 0.55 0.57 | 0.88 0.77 | 0.60 0.62 | 0.42 0.86 | 0.49 0.88 | 0.73 0.83 | 0.47 0.85 |
| PageRank (0.99) | 0.79 0.78 | 0.45 0.76 | 0.73 0.79 | 0.50 0.72 | 0.50 0.52 | 0.83 0.79 | 0.57 0.63 | 0.40 0.87 | 0.46 0.88 | 0.72 0.83 | 0.45 0.85 |
| Onion | 0.71 0.78 | 0.34 0.75 | 0.51 0.69 | 0.22 0.53 | 0.40 0.45 | 0.92 0.74 | 0.50 0.44 | 0.24 0.82 | 0.35 0.88 | 0.45 0.79 | 0.35 0.83 |
| Degree | 0.76 0.75 | 0.52 0.77 | 0.76 0.75 | 0.50 0.70 | 0.70 0.60 | 0.96 0.75 | 0.73 0.60 | 0.52 0.84 | 0.61 0.85 | 0.78 0.82 | 0.57 0.82 |
| RN-Rank | 0.83 0.70 | 0.68 0.75 | 0.84 0.78 | 0.56 0.72 | 0.72 0.57 | 0.85 0.76 | 0.89 0.60 | 0.61 0.84 | 0.69 0.83 | 0.87 0.80 | 0.61 0.81 |
| N2-Rank | 0.85 0.74 | 0.50 0.78 | 0.74 0.80 | 0.52 0.74 | 0.72 0.60 | 0.82 0.77 | 0.87 0.64 | 0.51 0.85 | 0.63 0.86 | 0.85 0.82 | 0.56 0.83 |

Table 5: Preservation ratio and NMI of Louvain of top $20\%$ ranked points

| Community # | 0 | 1 | 2 | 3 | 4 | 5 | 6 | 7 | 8 | 9 |
|---|---|---|---|---|---|---|---|---|---|---|
| $CC_G(V_i)$ | -0.14 | 0.07 | 0.2 | -0.15 | -0.09 | 0.04 | 0.02 | -0.2 | -0.3 | -0.26 |
| $CC_G(V_i \cap S)$ | 0.1 | 0.05 | 0.35 | 0.05 | 0.19 | 0.5 | 0.15 | -0.06 | 0.4 | -0.1 |

Table 6: A comparison of the concentration of the entire community vs. the subset of each community that is on the top $20\%$ ranked node by RN-Rank (denoted by $S$) on the T-cell dataset (Savas et al., 2018). Note that the concentration of the subset is strictly better for each community, denoting a presence of core within each community as per our MCPC definition

**Evidence of MCPC structure in single-cell data**    : As we briefly discussed in Section 4.1, we see some interesting evidence of presence of cores within communities of the single-cell data as per our definition.

Our large scale experiments on single-cell data has already established

i) As we select top ranked points, they are more separable into their ground truth communities.

ii) Our relative-centrality based approach results in significantly more balanced ranking.

While both of these observation mimic our simulation and theoretical observation in the MCPC structure, this does not directly indicate the presence of separable cores within each community in the real-world graphs. To test this, we calculate the concentration (as per Definition 2.1 in Section 3) of the nodes in each community and also the nodes of each community that are present in the top $20\%$ of the ranking. We use the T-cell data as an example. The values are noted in Table 6.

As we can see, the concentrations of the subsets are indeed consistently higher! This indicates that the points we select based on our ranking indeed leads a slight MCPC structure on the real graphs, making our theoretical insight on the unbalancedness of existing centrality measures in the MCPC structure more relevant.

Figure 9: Performance of the HITS+K-Means algorithm in (Elliott et al., 2020) when applied to balanced MCPC-block model

# E   DISCUSSIONS

## E.1   CORE DETECTION ALGORITHMS IN SINGLE-CORE PERIPHERY STRUCTURE

As we discussed, there exists a large literature of core-detection algorithms particularly focused on single core-periphery structure (Rombach et al., 2014; 2017; Yanchenko & Sengupta, 2023). In this direction, the recent and comprehensive survey (Yanchenko & Sengupta, 2023) cited that most of the core-periphery detection algorithms need $|E|^2 \log |V|$ or more time and highlighted centrality measures as being efficient. This, along with the ease of applying these methods to a multi-core periphery structure, motivated our choice of baseline.

## E.2   RESEARCH ON EXISTENCE OF MULTIPLE CORES

As we discussed, (Elliott et al., 2020) seems to be the only notable work in the literature that considered directed graphs with multiple cores. In this direction, we apply their core-detection algorithm to our MCPC-block model. Their method requires the knowledge of the number of cores and the number of peripheries. When the graph consists of 2 cores and 2 peripheries (as in our block model), their method involves first obtaining a 4-dimensional score using the popular HITS algorithm. Then, they apply $K$-Means with $K = 4$ to the four-dimensional dataset to separate into 4 blocks.

In this direction, we apply their method to a graph generated with the block probabilities in Table 1 with $\gamma = 0$ and $n = 4000$. Note that this is the simplest setting, where both cores have identical behavior in terms of core concentration, inter-core edges as well as overall community structure. We show the outcome in Figure 9. As we can observe, the output is not close to the ground truth. This can, in part, be attributed to the fact that their structure, although a multi-core directed one, is quite different from our MCPC structure.

Finally, (Tunç & Verma, 2015) seems to be the closest structure to our MCPC structure, even though it is in an undirected graph setting. Indeed, this seems to be one of the initiating works in aiming to understand the coexistence of community and core-periphery structure in a systematic manner, and is thus an important contribution. Their approach is as follows.

They also assume that the graph is partitioned into some ground truth communities, and each community has some core and some periphery vertices. They assume a hypothesis model in which the probability of an edge between $v_i$ and $v_j$ is $a\delta_{i,j}(C_i + C_j) + b$. We now decompress this definition. Here $\delta_{i,j} = 1$ iff $i$ and $j$ belong to the same community, and 0 otherwise. Next, $C_i = 1$ if $v_i$ is a core vertex. That is, i) All inter-community edge probability is a fixed $b$, ii) Intra-community core-periphery edge probability is $a + b$, and iii) Intra-core edge probability is $2a + b$.

Despite its expressibility, this structure has a few significant shortcomings.

First, it does not consider that the edge density can differ for different cores. We quantify this phenomenon with the concentration of a set of vertices and show that when cores have different concentrations, it can lead to many core ranking algorithms performing in an unbalanced manner, which we mitigate with our novel relative centrality framework.

Next, the model also does not consider that the inter-community edge probability between core vertices is less than between peripheral vertices. We capture this in our $(\alpha, \beta) - \text{MCPC}$ structure definition, and this observation allows us to obtain subsets of real-world datasets with better community structure by using our core ranking algorithms.

Finally, they do not present any core detection algorithm beyond a maximum likelihood approach w.r.t. the inference model we discussed. It is well known that such methods may have very slow convergence. Indeed, the experiments in (Tunç & Verma, 2015) consider graphs with less than 200 vertices. In comparison, we apply our methods to datasets with $> 50,000$ points and terminate generally in less than 10 seconds, owing to our fast $\mathcal{O}(|E| + |V| \log |V|)$ run time.

### E.3 LIMITATIONS AND FUTURE DIRECTIONS

In this paper, we take a step toward quantification of the MCPC structure, and design of core-ranking algorithms that mitigate unbalancedness observed in traditional centrality measures when applied to MCPC structures. In this direction, there are several questions/directions that we are unable to explore in this paper, which can be thought of as our limitation. We place them here.

1. Our proofs are in the MCPC-block model, which we use to prove the unbalancedness of traditional centrality measures and the performance of our algorithm. It will be interesting to obtain similar proofs in the other simulation model we study, the concentric GMM model. Furthermore, our proofs can be extended to the case when the size of the cores and peripheries are different.

2. We proposed the algorithm concept of relative centrality and proposed the meta-algorithm Meta-Rank in Algorithm 2. We used two instantiations of our algorithm, namely N2-Rank and RN-Rank and showed interesting ICEF improvement vs. balancedness tradeoffs. However, obtaining the best possible instantiations for different problems remains an interesting open direction.

3. Even though our methods provide a superior preservation ratio compared to the traditional centrality measures in the single-cell data, we also have a lower-than-ideal preservation ratio in some datasets, such as the Segerstolpe dataset. Designing algorithms with even higher balancedness is an important goal.

4. In this paper, our real-world experiments focused on a large set of single-cell datasets. We believe MCPC structures may be present in other kinds of datasets as well. Exploring the existence of such structures in different domains is also important.

5. We show a concrete application to single-cell datasets whereby the subsets are better separable into their ground truth communities. We chose the top $20\%$ of the points for the experiments. Overall, the higher the number of points we select from this ranking, the higher the preservation ratio (less unbalancedness), but the lower the improvement in separability. As in real-world graphs it is not expected that there is a clear separation between core and periphery vertex behavior within communities, a clear cutoff choice is not apparent. However, we observe that choosing any cutoff percentage between 20 and 40 results in a noticeable improvement in the clustering accuracy while maintaining a good preservation ratio.

6. Although we have *significantly* higher balancedness/preservation that all baseline centrality measures, the NMI/purity improvement of clustering on the points selected by our methods is lower, even though the improvements are *comparable*. However, such a tradeoff is inevitable. If the selected points have very low balancedness, then it indicates that only a few points from some communities are present (some communities can be also completely missing) and therefore the total number of inter-community edges go down, resulting in a *visibly* better clustering outcome (at the cost of losing information about many underlying communities). Analogous phenomena have been widely studied in the "quality-fairness" tradeoff in case of the supervised fair-clustering paradigm (Hakim et al., 2024). Studying the limit of the "clustering improvement-balancedness tradeoff" in our unsupervised ranking scenario is a very interesting and important future direction.

7. Finally, we note that in this paper, we used our methods to obtain a subset of the dataset that is better separable into the ground truth communities. An important next step, and thus

a limitation of this work, is that we do not use the clustering on this set to obtain a better clustering of the whole dataset. While we have some progress in this direction, it is beyond the scope of the paper and requires a systematic study.

# F    PLOTS OF ICEF OF INDUCED SUBGRAPH AND PRESERVATION RATIO OF ALL SINGLE-CELL DATASETS

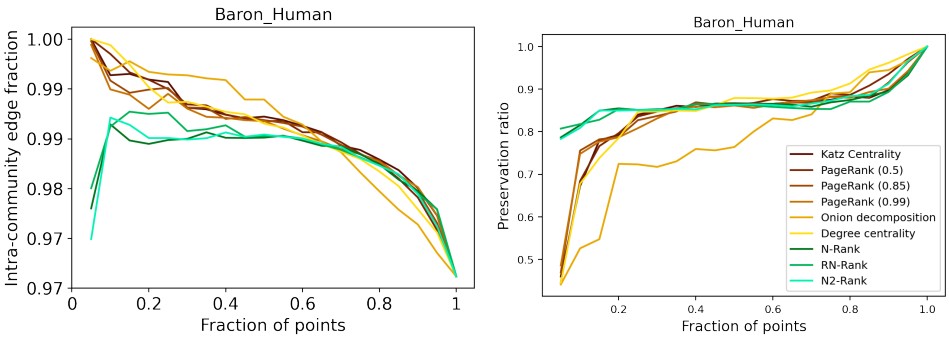

Figure 10: Baron Human dataset

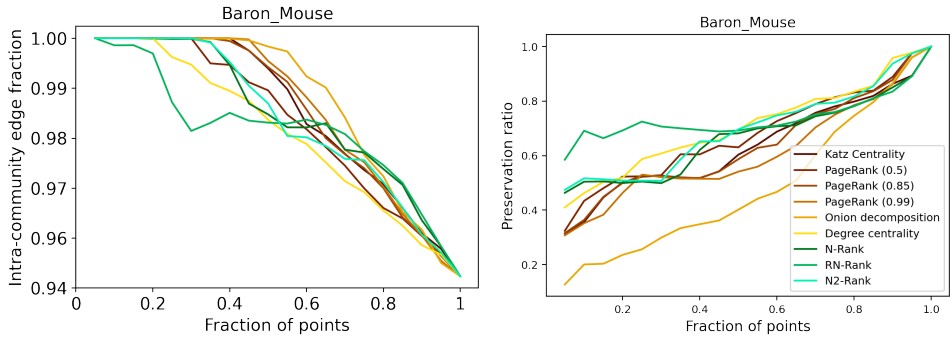

Figure 11: Baron Mouse dataset

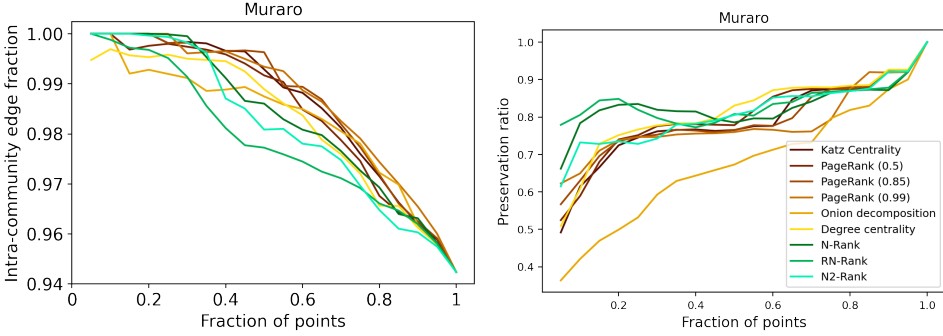

Figure 12: Muraro dataset

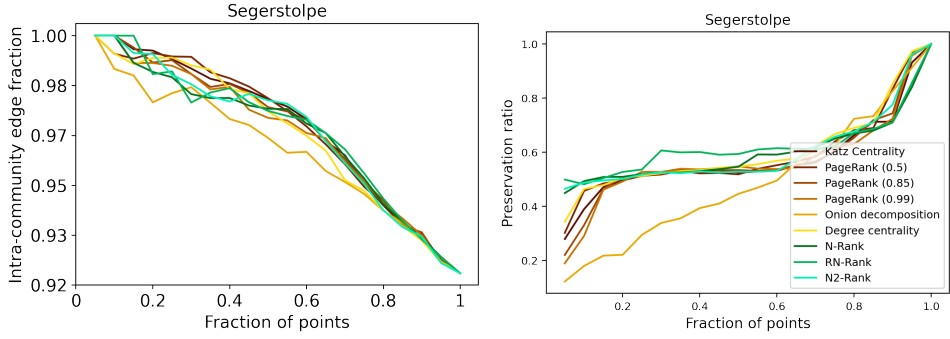

Figure 13: Segerstolpe dataset

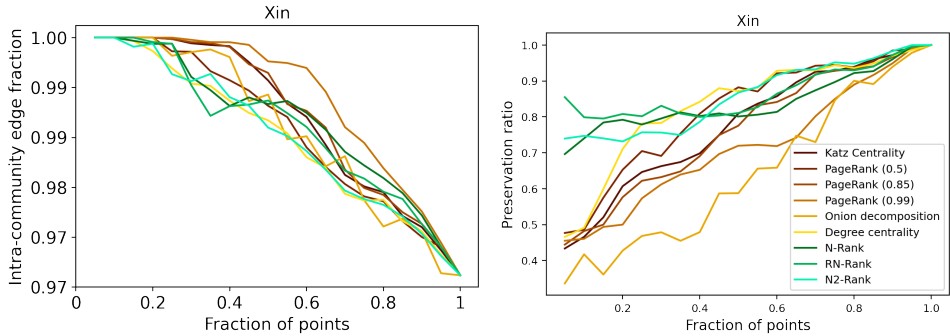

Figure 14: Xin dataset

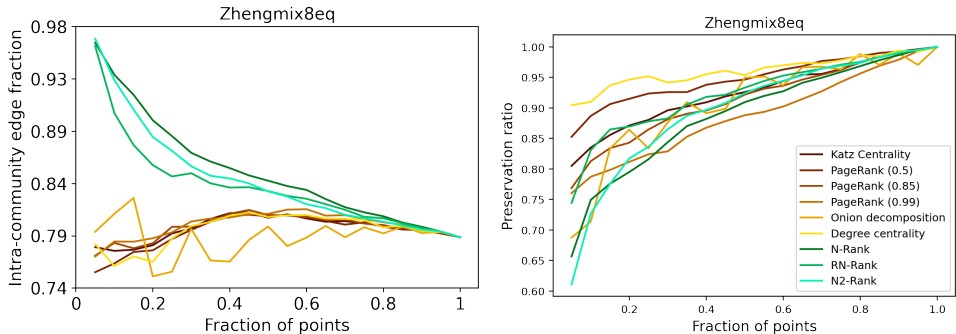

Figure 15: Zhengmix8eq dataset

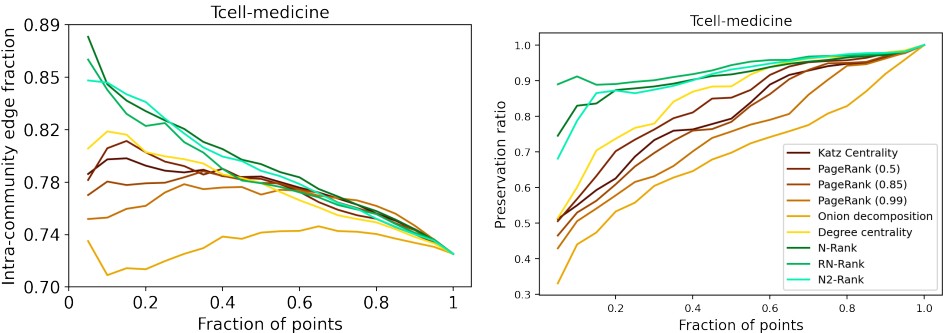

Figure 16: Tcell dataset

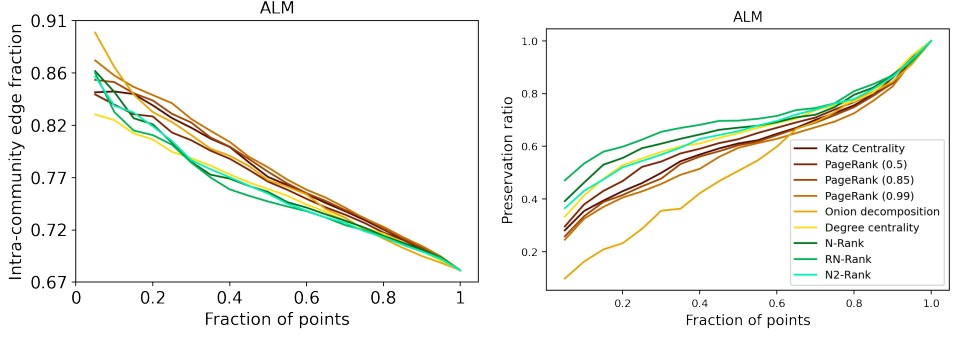

Figure 17: ALM dataset

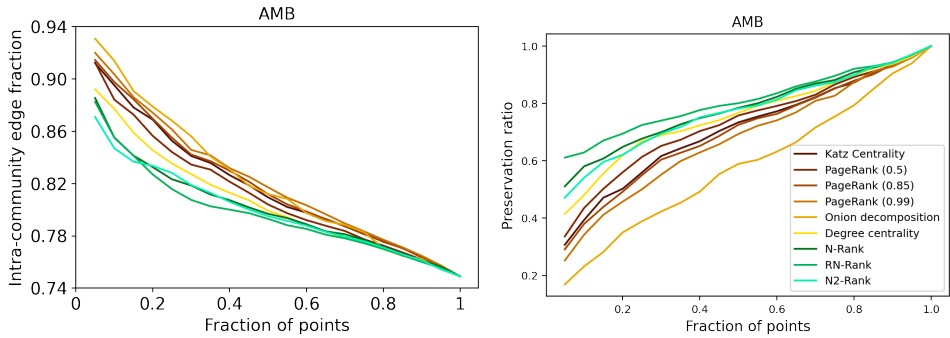

Figure 18: AMB dataset

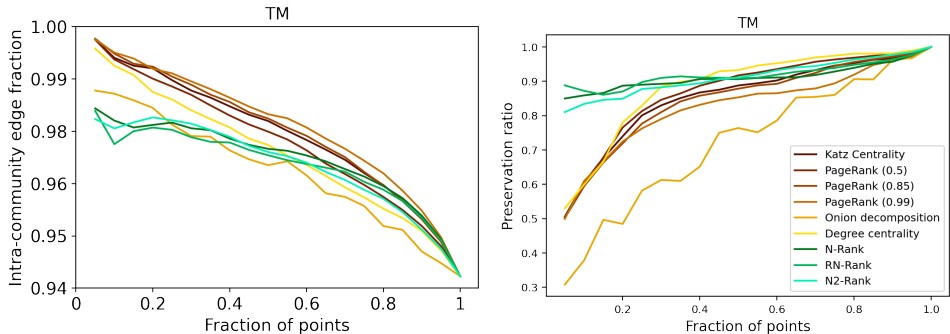

Figure 19: TM dataset

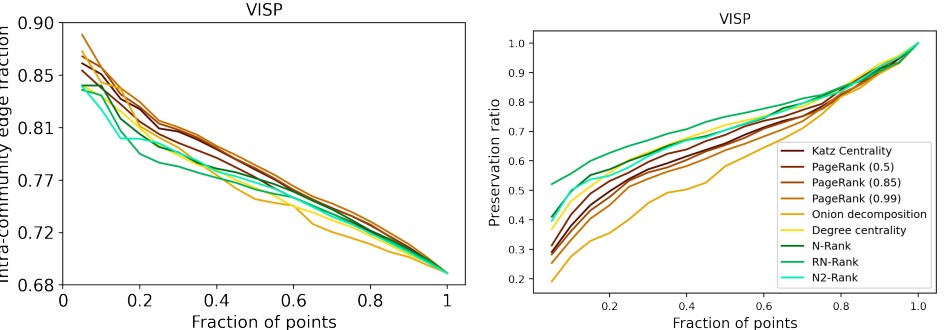

Figure 20: VISP dataset

