# OpenReview forum: "Balanced Ranking with Relative Centrality: A multi-core periphery perspective"
_ICLR.cc/2025/Conference — ICLR 2025 Poster_

### Official Review · Reviewer_6naw · 2024-11-01

**Soundness:** 2
**Presentation:** 3
**Contribution:** 3
**Rating:** 6
**Confidence:** 3

**Summary:**

This paper focuses on the task of **unsupervised ranking on graphs** and aims to generate **balanced ranking**, where the top-ranked nodes contain a reasonable fraction of nodes from each community on the graph.

The authors propose a novel notion called **relative centrality**, which better preserves balancedness across different communities. Based on relative centrality, the authors propose several new approaches to iteratively update the centrality scores, which can be subsequently used for node ranking and graph clustering.

On the other hand, the authors propose a novel structural assumption for the underlying graphs, called **multi-core-periphery with communities (MCPC)**. Based on this, the authors define a stochastic block model and show that typical centralities are unbalanced under this model. Finally, experiments on 11 single-cell datasets are conducted to show that the proposed methods achieve higher balancedness while maintaining similar clustering quality.

**Strengths:**

1. The goal of computing balanced ranking for vertices on graphs is a meaningful and interesting problem.
2. The proposed assumption of multi-core-periphery structure is a natural combination of the community structure and the core-periphery structure, and the intuitions are conveyed nicely through Figures 2 and 3.
3. The proposed methods and conducted experiments are generally described in detail.

**Weaknesses:**

1. I am not sure if this work is interesting to the ICLR community. This paper deals with unsupervised ranking, which can be regarded as unsupervised learning, but it may not fall into the scope of "representation learning," which is "generally referred to as deep learning" as indicated on the official website of ICLR 2025. This paper may be more suitable for some venues on data mining or network science, for example.
2. The theoretical analysis seems limited and not supportive for the balancedness of the proposed methods. The analysis relies on several simplifications: for example, the number of underlying communities is $2$, the size of all the cores and peripheries are the same, and $t=1$ in Theorem 3.6. The authors do claim that the analysis can be extended, but do not provide further explanations. On the other hand, although Theorem 3.6 and Lemma A.7 verify that the relative centrality scores of core vertices are close to $1$ and larger than those of periphery vertices, this does not imply that the induced ranking is balanced, since the scores in one community may be all larger than those in other communities.
3. The paper is messy and needs significant improvement in presentation and layout. First, there lacks a detailed section on related work, making it hard to judge the contribution of this paper and its relevance to the ICLR community. Although Section 2.1 discusses some related work, it is brief and only concerns part of the contributions of the paper. Second, the text for the proposed assumption, methods, and analysis are not structured nicely, which is somewhat confusing. In particular, the order of the main text is not consistent with the contributions outlined in Section 1.1. Finally, there are numerous writing issues that affect the readability of the paper, as listed below.
4. Some background concepts are not explained clearly. For example, the meaning of the single-cell data, the metrics of NMI and purity, and the "onion" baseline are not introduced clearly enough.
5. The experiments only focus on single-cell datasets, which is limited. More experiments on other types of networks (e.g., social networks) are expected, and the number of tested single-cell datasets can be reduced.

Minor issues:

1. Most (if not all) citations in this paper should use `\citep{}` instead of `\cite{}`, so that the author names are placed in the parentheses. Also, the authors should cite the published version instead of the arXiv version of some papers.
2. Line 163: "community structure" -> "core-periphery structure".
3. Line 287: here the notation $k$ is ambiguous since it has a different meaning in Line 286.
4. Line 352: "$N_{G}(v_{j})$" -> "$N_{G}(v_{i})$". Also, here the term "neighborhood" should be specified as "in-neighborhood" or "out-neighborhood".
5. Line 762: "upper bounded" -> "lower bounded".
6. There are some grammatical issues or typos:
	1. Lines 22-23;
	2. Line 74: "for e.g.," -> "e.g.,";
	3. Line 130: "behind of" -> "behind";
	4. Line 180 and other occurrences: "w.r.t" -> "w.r.t.";
	5. Line 264: remove "is defined";
	6. Lines 419 and 1012: remove repetition of "look at";
	7. Line 463: remove "to";
	8. Lines 75 and 270: add space before left parentheses;
	9. Lines 192-193: the parentheses are not matched;
	10. the hyphens in compound words should be used correctly and consistently. For example, it should be "centrality-measure-based" in Line 17 and "Core-Periphery structure" in the caption of Figure 2(b).
7. I recommend to beautify the layout of the paper:
	1. the captions of Figure 4 are hard to read;
	2. the annotation in Lines 352-353 are separated across lines;
	3. Table 1 and Figure 5 are placed weirdly;
	4. there should be punctuations around multiline math expressions;
	5. math expressions are not aligned nicely, and the ones at the top of page 15 are aligned terribly;
	6. the font of notations in math expressions is inconsistent in many places (e.g., $k$, $\mathrm{deg}(\cdot)$, and $o(\cdot)$ in Lines 849-856);
	7. the text font changes in Lines 1332-1346.

**Questions:**

1. Could you provide more justifications for the relevance of this work to the ICLR community? For example, you can give some relevant papers published in ICLR or similar conferences/journals and add discussions to the paper.
2. The quantities in Table 1 are magical to me. Could you explain them?
3. Could you explain "single-cell RNA seq data" in more detail? Why do you focus on directed graphs as stated in Line 80? What about for undirected graphs?

---

> ### Author Response · Authors · 2024-11-16
>
> We thank the reviewer for their review and comments. We shall fix the minor issues pointed out by the reviewer in the revised version of the paper. Please find our response to the other weaknesses and the questions below.
>
> ## Response to weaknesses:
>
> We have responded to the reviewer's concern regarding the relevance of our paper in the general comment at the top of the page and as an answer to their Question 1 below.
>
> ---
>
>
> **Comments on the theoretical results:**  We primarily focused on theoretically proving the unbalancedness of traditional centrality measures and only initial evidence of the change in the behavior of relative-centrality-based algorithms compared to traditional centrality measures. We aim to strengthen the proof for N-Rank further and show exact balancedness instead of only arguing that points from a weaker core have a higher score in a future journal version of the work, along with further generalization of the setup (k>2, different sizes of communities, etc.). In fact, we focused on the case where the size of both communities (and both the cores and peripheries) are the same to show that unbalancedness in traditional centrality measures can occur even when the communities are *balanced in size*.
>
> ---
>
> **Background concepts:** To provide the reviewers with more details on single-cell data, we describe their structure, importance, and relevance to the ICLR community in the general comment at the top of the paper. We shall also describe the clustering metrics NMI and purity in the appendix and add more description of the onion baseline, which we have cited as a core-decomposition-algorithm in the "baseline centrality measures" paragraph. We will explicitly mention that this is the onion baseline. We apologize for the confusion.
>
> ---
>
> **Using several single-cell datasets:** The reviewer suggests using other types of datasets and reducing the number of single-cell datasets. We have already described our motivation for using single-cell data for our experiments. Here, we also comment on why experiments on more single-cell data can contribute positively.
>
> In single-cell analysis, the data obtained suffers from different technical and experimental noises. This can affect the structure of the graph
> embeddings for different datasets. In our experiments, we use datasets from many different sources, and both the higher preservation ratio of our ranking algorithms and the improved performance of clustering algorithms on the top-ranked points across all the datasets give us more confidence in the applicability of our algorithm to future datasets. Furthermore, we presented our results on all of the datasets to maintain transparency as much as possible.
>
>
> ---
> ---
>
> ## Answer to questions:
>
> **On the relevance of our work to the ICLR community:**
>
> We have written a detailed general audience comment on why single-cell analysis is relevant to the ICLR community at the top of this page. Here, we give more evidence of the relevance of our paper in this community.
>
> First, we note that unsupervised learning seems to be a relevant topic to the ICLR community. As a recent and relevant evidence, we cite the paper [3], an oral paper in ICLR 2024 that focused on fast algorithms for K-Means with guarantees of statistical optimality.
>
> Secondly, besides using ranking as a preprocessing of the clustering algorithm, we can also use the ranking algorithm as a preprocessing step of deep learning algorithms. It was recently noted that the zero-shot learning performance of foundational models in genomics data is lacking [4]. It will be very interesting to explore if the performance of these models can be improved when only considering the cores of a dataset. Furthermore, our identification of peripheries (harder-to-cluster points) should also have applications in contrastive learning in the context of "hard negative samples" [5].
>
> ---
>
> **More explanation on single-cell data:** We request the reviewer to read our general comment on our motivation at the top of the page, which discusses the structure, importance, and relevance of single-cell RNA seq data to the ICLR community in detail.
>
> ---
> ---
> [3] ``Statistically Optimal K-means Clustering via Nonnegative Low-rank Semidefinite Programming.'', Yubo Zhuang, Xiaohui Chen, Yun Yang,  and Richard Y. Zhang, ICLR 2024 (oral). https://iclr.cc/virtual/2024/oral/19717
>
> [4] ``Assessing the limits of zero-shot foundation models in single-cell biology'',  Kasia Z. Kedzierska, Lorin Crawford, VAva P. Amini, and Alex X. Lu, BiorXiv:  https://doi.org/10.1101/2023.10.16.561085
>
> [5] ``Contrastive Learning with Hard Negative Samples'', Joshua Robinson, Ching-Yao Chuang, Suvrit Sra, Stefanie Jegelka, ICLR 2021. https://arxiv.org/pdf/2010.04592

---

> ### Author Response · Authors · 2024-11-16
> **Continued response:**
>
> Finally, we explain the quantities of Table 1.
>
> **Explanation of Table 1:**  Table 1 shows the block probability parameters we chose for generating the simulation graphs. Note that the vertices in the graphs are divided into community $V_0$ and $V_1$ (Each $V_i$ is further separated into a core $V_{i,1}$ and the rest being periphery nodes $V_{i,0}$). Then, for any two vertices $u \in V_{i_1,j_1}$ and $v \in V_{i_2,j_2}$ the directed edge $u \rightarrow v$ is added to the graph with probability as indicated in the $V[i_1,j_1]$-th row and $V[i_2,j_2]$-th column of Table 1. We use different choices of $\gamma$ to generate different graphs. For example, if we set $\gamma=0$, the probability of an edge between any two core vertices of the same community is 0.8. In contrast, the probability that an edge exists between two core vertices from the different communities is $0.05$.
>
> We chose the values in Table 1 in such a way that:
>
> 1) For any value of $0 \le \gamma \le 0.1$ the graph exhibits an MCPC structure with respect to the underlying cores and peripheries.
>
> 2) As $\gamma$ increases, the cores become more unbalanced, which results in the performance of traditional centrality measures becoming more unbalanced (as observed through the simulations).
>
> This setup allows us to study the balancedness of our algorithm and the unbalancedness of traditional centrality measures systematically via extensive simulations.
>
> ---
> ---
>
> We thank the reviewer again for their efforts and look forward to answering other questions that they may have.

---

> > ### Comment · Reviewer_6naw · 2024-11-18
> > **Response to the Authors' Rebuttal**
> >
> > Thank you for your response. The background and motivation behind the paper's concentration on single-cell data are now reasonable to me.
> >
> > Now, my major concern remains in the paper's (i) theoretical contributions and (ii) presentation.
> >
> > (i) If the authors admit that "We primarily focused on theoretically proving the unbalancedness of traditional centrality measures and only initial evidence of the change in the behavior of relative-centrality-based algorithms compared to traditional centrality measures," then some claims seem to be overclaimed and misleading, such as "1-step N-Rank is good" in Line 399 and "ranking the vertices in the descending order of $\hat{F}$ gives a balanced ranking with high core prioritization" in Lines 405-406. These claims may cause the readers to overestimate the theoretical contributions. Instead, these phenomena can be reported as empirical observations.
> >
> > (ii) The presentation can and should be improved. The authors' explanations given in the general comment should be added to the paper, and Section 2.1 should be moved and extended to a "related work" part. Also, it may be better to move the detailed definition of MCPC (now at the beginning of Section 3) to Section 2 and move the simulation results to Section 4, thus focusing Section 3 on the analysis of centrality and relative centrality as its header indicates. You can also consider modifying the overall structure otherwise. The current presentation quality is clearly below the acceptance bar from my perspective, and I will consider raising my score if the revised version is fairly satisfying.
> >
> > Another minor comment: I recommend to standardize the format of the bold headings in the main text.

---

> > > ### Author Response · Authors · 2024-11-26
> > >
> > > We thank the reviewer for their helpful suggestions and apologize for the delayed response. The reviewer had two primary concerns, namely, our theoretical contribution (and ensuring that we do not oversell) and our presentation. Following their comments, we have revised our paper (and uploaded it), which contains the following changes.
> > >
> > > 1) **Theoretical contribution:** We agree with the reviewer that our previous Theorem on N-Rank only shows that all core points have high value but do not give any direct evidence of balancedness. In this direction, we have added a new Theorem (Theorem 3.3) which shows that on expectation, around $\frac{(1 \pm \epsilon)}{\mathbb{P}[(\ell,1)(\ell,1)}$ fraction of points from each core get an N-Rank score of $1$ (which is the maximum possible score). The proof is placed in Appendix A.4. This gives more concrete evidence that the top of the ranking is balanced. We have still modified the writing to ensure we do not oversell our theoretical results. Making these results more robust (such as converting the expectation to a high probability bound and proving balancedness among core points with lower scores) is a future theoretically centered direction.
> > >
> > > ---
> > >
> > > 2) **Presentation:** We thank the reviewer for their very useful remarks on the presentation of the paper. We have gone through all the minor issues and corrected them throughout the paper. Among the more visible changes are
> > > i) We have moved the formal description of the MCPC structure to Section 2. Section 3 now only deals with the random graph model (both theorems and supporting simulations).
> > > ii) We have added more discussion on single-cell data on Page 2 (line 68 onwards) and again on Page 5 (line 262 onwards).
> > > iii) We have moved the related works section to the end of the Introduction.
> > > iv) We have fixed the orientation of Table 1, and Figure 5 and the caption of Figure 4.
> > > v) We have corrected the notational inconsistencies throughout the paper.
> > >
> > > ---
> > > ---
> > >
> > > We thank the reviewer again for their valuable comments and look forward to answering further questions.

---

> > > > ### Comment · Reviewer_6naw · 2024-11-27
> > > > **Response and Score Raising**
> > > >
> > > > Thank you for your response and careful revision of the paper.
> > > >
> > > > I have read through the revised paper as well as your discussions with other reviewers, and I find that my concerns have been mostly solved. Now the paper states the contributions clearly with explicit explanations and emphasis on single-cell datasets, and the theoretical results have been strengthened and claimed objectively. On the other hand, I understand that the paper does an innovative job of pointing out the rationale behind previous methods' unbalancedness and initializing new relative-centrality-based ranking methods. Thus, I think the purity-balancedness tradeoff is not a major problem for this paper.
> > > >
> > > > However, I find Theorem 3.3 technically weird, as it states that the expected number of nodes in question lies in the range of $(1\pm\epsilon)$ times some fixed amount for any $\epsilon>0$. Mathematically, this implies that the two amounts are equal and does not make sense. I believe that the application of Chernoff bound in Lines 1112-1118 yields some constraints on $\epsilon$ depending on the expectation value and $n$. Additionally, I admit that I do not have time to check the math parts carefully and am not sure if they are reasonable, but also I am not taking this as a key judging factor for this type of ICLR submission.
> > > >
> > > > Overall, I decide to raise my rating from 3 to 6.
> > > >
> > > > ---
> > > >
> > > > Below are some (very) minor comments in case you need them in the future (they don't affect my current rating):
> > > >
> > > > 1. Some of my previous comments have not been solved completely, e.g., the citations in Line 143 and punctuations around math;
> > > > 2. Line 121: remove 2nd occurrence of "in Section";
> > > > 3. Line 395 and other occurrences: "on expectation" -> "in expectation";
> > > > 4. Lines 400-402: remove "to" and repetition of "any";
> > > > 5. Line 430: "Large scale" -> "Large-scale";
> > > > 6. Line 483: are the two "CC"'s supposed to be in different fonts?
> > > > 7. Line 1157: the lower bound in the integral should be $-\infty$;
> > > > 8. Lines 1163-1167: it seems that these two lines of derivation only switch the ordering of the factors and it should be equality?
> > > > 9. Lines 1194-1201: the definition of the events are not standard. You can write "define $E_1$ to be the event that '...'";
> > > > 10. Lines 1404-1405: change "T" to `$t$` if you mean it.

---

> > > > > ### Author Response · Authors · 2024-11-28
> > > > >
> > > > > We thank the reviewer for recognizing the strengths of the paper. We also thank them for their very valuable suggestions. We believe it has noticeably improved the presentation quality of our paper. We shall also incorporate the minor suggestions.
> > > > >
> > > > > Regarding the new theorem, the reviewer is correct. Essentially, the probability bounds due to the Chernoff and other analytical tools we use lead to a multiplicative $1 \pm o(1)$  error with $n$, which we wrote down in terms of absolute constants. The correct statement should be as follows.
> > > > >
> > > > > Given a block probability matrix $\mathbb{P}$, the following happens. For any $\epsilon>0$, there exists  $n_{\epsilon}$ such that if an MCPC block model graph is generated on $n>n_{\epsilon}$ vertices using $\mathbb{P}$ (and $k=\omega(\log n)$) then the expected fraction of vertices with score $1$ from any core $V_{\ell,1}$ lie within $(1 \pm \epsilon) \cdot \frac{1}{\mathbb{P}[(\ell,1),(\ell,1)] \cdot k }$.
> > > > >
> > > > > We shall fix the statement of the theorem to address this or otherwise simplify the statement (such as directly writing the bounds in terms of the multiplicative $(1 \pm o(1))$ error).
> > > > >
> > > > > Again, we thank the reviewer for their effort and will gladly address any other questions/comments they may have.

---

### Official Review · Reviewer_mu1B · 2024-11-03

**Soundness:** 3
**Presentation:** 3
**Contribution:** 2
**Rating:** 5
**Confidence:** 5

**Summary:**

The paper argues against global centrality measures such as PageRank for ranking nodes and suggests using relative centrality instead. As the name suggests, relative centrality measures centrality of a node relative to its neighborhood. The paper shows that relative centrality on Louvain community detection algorithm produces better clusters (as measured by preservation ratio of top 20% points and purity score).

**Strengths:**

The paper has a limitations section. Kudos to the authors for being honest about the technical limitations of their study.

**Weaknesses:**

- I have no objection to adding another centrality measure to the long list of node centrality measures. However, the results would have been more convincing if the experiments had been conducted for recommender systems rather than for community detection.

- The authors may find these references related to their work:

Sotiris Tsioutsiouliklis, Evaggelia Pitoura, Panayiotis Tsaparas, Ilias Kleftakis, and Nikos Mamoulis. 2021. Fairness-Aware PageRank. In Proceedings of the Web Conference, pp. 3815–3826. https://doi.org/10.1145/3442381.3450065

Kijung Shin, Tina Eliassi-Rad, Christos Faloutsos. 2016. CoreScope: Graph Mining Using k-Core Analysis - Patterns, Anomalies and Algorithms. In Proceedings of the IEEE International Conference on Data Mining, pp. 469-478. https://ieeexplore.ieee.org/document/7837871

- The captions for Figures 10 to 20 should be more informative. As is, they only list the name of the dataset.

**Questions:**

Ranking is often used in recommender systems. The authors point this out in the first sentence of the introduction. Why did they not compare relative centrality for recommending nodes instead of using it for community detection?

---

> ### Author Response · Authors · 2024-11-16
>
> We thank the reviewer for their comments. Please find our responses to the mentioned weaknesses and questions below.
>
> ## Response to weaknesses:
>
> **Comments on the usefulness of our algorithm:** While we agree that a long list of centrality measures exists, the analysis of balance in centrality measures is not a well-explored topic. In this direction, we provide a formal setup to analyze the balancedness of ranking algorithm and design a class of simple and fast balanced ranking algorithms. After a comprehensive survey, we believe that we are the first paper to observe the unbalancedness of popular ranking centrality measures such as Pagerank in a formal setting.
> As the notion of balancedness is inter-coupled with the presence of underlying communities, we focus on a concrete application, i.e., improving the clustering of single-cell data, which we think is an important question in genomics that is gaining attraction from the ML community. We request the reviewer to read our general audience comment at the top of the page for more motivation.
>
> ---
>
> **Response to the shared papers:** We thank the reviewer for sharing the papers. Below are our responses based on a first read-through.
>
>
> We note that fairness-aware PageRank is another example of a *supervised* balanced ranking algorithm focused on obtaining a PageRank-like outcome such that the total value of a specific and defined set of nodes is more than some given value. Our setting is the harder "unsupervised" ranking problem, where the group identities of the points are unknown. In fact, we show that our algorithm can produce such a ranking that identifies the core points from all the communities and shows the impact of such ranking on the very important ``clustering of single-cell data'' problem.
>
>
> The other paper uses $k$-core properties of social network graphs to solve various problems. While they do talk about core-periphery and community structure, in our understanding, the structure it explores differs from the ``coexistence'' of the core-periphery and community structures, and it does not focus on *balanced* ranking, which is a main contribution of our work.
>
> ---
> ---
>
> ## Answer to the question regarding our choice of data:
>
> We were motivated to study this problem by trying to improve the clustering performance of algorithms on single-cell data. However, as we do not use any domain knowledge of this datatype, we believe our method can be applied as a general ranking algorithm.
> We point the reviewer to our general audience comment at the top of the page to get a more in-depth view of our motivations for focusing on single-cell data. We agree that the performance of our algorithms in other domains (including recommender systems) should be future areas of exploration.
>
> ---
> ---
>
> We thank the reviewer again for their efforts and will be happy to answer any other queries they may have.

---

> > ### Comment · Reviewer_mu1B · 2024-11-25
> >
> > Thank you for your feedback.
> >
> > I doubt that experiments on community detection (which satisfies the no-free-lunch theorem [1]) on single-cell data translate to the wider applicability as claimed. Biological and genomic data are notoriously different than say social data [2][3].
> >
> > In terms of supervised vs. unsupervised, one can easily detect groups in social networks [4].
> >
> > References
> >
> > [1] Leto Peel et al. ,The ground truth about metadata and community detection in networks. Science Advances 3, e1602548 (2017). https://www.science.org/doi/10.1126/sciadv.1602548
> >
> > [2] Gabriel Budel and Maksim Kitsak. Complementarity in Complex Networks.  arXiv:2003.06665v2, March 2023. https://arxiv.org/abs/2003.06665
> >
> > [3] Kovács, I.A., Luck, K., Spirohn, K. et al. Network-based prediction of protein interactions. Nat Commun 10, 1240 (2019). https://doi.org/10.1038/s41467-019-09177-y
> >
> > [4] David Liu et al. Group fairness without demographics using social networks. FAccT'23. https://arxiv.org/abs/2305.11361

---

> ### Comment · Area_Chair_4J4d · 2024-11-25
>
> Could please acknowledge and respond to the rebuttal.

---

> ### Author Response · Authors · 2024-11-26
>
> We thank the AC for ensuring an interactive discussion and thank the reviewer for their feedback.
>
> We are unsure about the relevance of the reviewer's latest comments. We never claim that Biological and genomic data have the same structures as social networks. We simply mentioned that some social networks may have an MCPC structure. We have re-emphasized in our earlier comments (and in our paper) that our focus is on single-cell data, which itself is a very important domain. Testing on other data types is a future direction.
>
> Regarding their comment on social networks, the reviewer said that "detecting groups in social networks is easy". This seems like an arbitrarily imprecise statement, as different kinds of social networks could have incomparable structures, and again, this is not the paper's focus.
>
> The experimental focus of our paper is on single-cell data, which is very important (as described in our general audience comment) and which we capture with our MCPC structure. In this context, we have provided a clean theoretical analysis of the unbalancedness of existing centrality measures and theoretical and experimental support for our superior performance.
>
> The reviewer has continued to talk about social networks throughout the discussion, which is not the paper's focus (we have only pointed out that MCPC could be applicable in some social choice scenarios as core-periphery and community structure could coexist in "some" social networks). Therefore, we are unsure about how to satisfy their concerns.
>
> We thank the reviewer for the discussion.

---

> > ### Comment · Reviewer_mu1B · 2024-11-26
> >
> > The reason I mentioned social networks is because of the following comments from the authors:
> >
> > In the authors comments from 16 Nov 2024 at 16:11 comments, they wrote: "We believe it can be applied to high-dimensional noisy data from other domains."
> >
> > In the author's "Answer to the question regarding our choice of data" of 16 Nov 2024 at 16:32pm, they wrote: "we believe our method can be applied as a general ranking algorithm."
> >
> > So, which is it? Is the proposed method "a general ranking algorithm" that "can be applied to high-dimensional noisy data from other domains"? Or is it not? If it is, then social network data is "high-dimensional noisy data from other domains".
> >
> > As I wrote in my original review: " I have no objection to adding another centrality measure to the long list of node centrality measures." However, it behooves us to be clear and concise about the claims we make in our papers.

---

> > > ### Author Response · Authors · 2024-11-26
> > >
> > > We thank the reviewer for their response.
> > >
> > > We completely agree with the reviewer that we should be clear and concise about the claims we make in the paper. First, we direct the reviewer to the exact wording of our contribution.
> > > "A Balanced meta-ranking-algorithm. As the primary contribution, we coin a novel concept, “relative centrality,” and design a meta-ranking algorithm (Details in Section 3.1) that provides superior balancedness to several popular centrality measures on the graph embeddings of a large set of biological (single-cell) datasets."
> > >
> > > That is, we have clearly mentioned that the experimental success of our paper is in single-cell data.
> > >
> > > ---
> > >
> > > Secondly, we summarize our reasons behind proposing this algorithm as a general-purpose algorithm (as opposed to an algorithm just for single-cell data).
> > >
> > > In *Appendix C*, we have shown that K-NN embeddings of a generalization of the famous Gaussian mixture model (GMM) show MCPC characteristics, and our algorithm has superior balancedness than traditional centrality measures in this setting as well. GMM is a widely used model in ML literature that is used to explain the behavior of different algorithms and data types (and is not specific to single-cell data). Furthermore, as we do not use domain knowledge of single-cell data (such as focusing on specific genes based on their well-known relevance), and our theoretical model of MCPC combines natural graph structures (community and core-periphery) that are present in many different domains, we present our algorithm as a general ranking algorithm that "could" have applications beyond single-cell data.
> > >
> > > ---
> > >
> > > **To be exact**, we believe our algorithm "could" have applications beyond single-cell data in "some" other domains (motivated by our observations of Appendix C in part). However, this in no way implies that our algorithm is successful in "every" high dimensional noisy dataset from "every" other domain.
> > >
> > > The latter seems to be the reviewer's interpretation, as they focus on random examples from a randomly chosen domain (social networks) to discuss our contributions. Understanding the impact of our algorithm in other domains beyond single-cell is a future step that we do not claim as a contribution in this paper. In fact, we do note this in the limitations of Appendix E. We have significant performance on a large set of single-cell data (along with theoretical and simulation support), which itself is an important application and forms the experimental support of our paper.
> > >
> > > We hope this addresses the reviewer's concerns.

---

> > > > ### Comment · Reviewer_mu1B · 2024-11-26
> > > >
> > > > I am not sure what the authors are suggesting by "reviewer's interpretation". I quoted the feedback provided by the authors:
> > > >
> > > > 16 Nov 2024 at 16:11: "We believe it can be applied to high-dimensional noisy data from other domains."
> > > >
> > > > 16 Nov 2024 at 16:32pm: "we believe our method can be applied as a general ranking algorithm."
> > > >
> > > > Also, sentences such as the following are not clear and concise:
> > > >
> > > > "we believe our algorithm "could" have applications beyond single-cell data in "some" other domains (motivated by our observations of Appendix C in part)."
> > > >
> > > > What are those "some other domains"? I strongly recommend removing such speculative sentences.
> > > >
> > > > That is all.

---

> ### Author Response · Authors · 2024-11-26
>
> The reviewer said, "However, it behooves us to be clear and concise about the claims we make in our papers." We agree with this.
>
> We note that we have not made any unsubstantiated claim in the *paper*. In the paper itself, we have clearly mentioned that we have focused on single-cell data and have claimed only that as our contribution (along with other theoretical and simulation counterparts), which we think is an important application.
>
> Earlier in our rebuttal, we mentioned, “Therefore, we propose our framework as a ranking algorithm, with the primary application explored in this paper being towards improving community detection. Furthermore, our ranking algorithm does not rely on domain knowledge of single-cell data. We believe it can be applied to high-dimensional noisy data from other domains.”
>
> To begin with, this quote constitutes a high-level discussion in the rebuttal. It is not as rigorous as the paper because the space is limited. Secondly, one should read the whole paragraph together rather than a single sentence. We mentioned that our algorithm didn’t use any domain knowledge; hence, this is a general algorithm. we can apply it to other domains. This is a factual statement. This does not mean that we claim our algorithm will perform well in other domains. We treat this as an open direction. Whether it performs well in other domains, such as social networks, would need rigorous experimentation, and it is a future direction that does not impact the paper's contributions. Continuing a discussion about these directions deviates from the focus of the paper.
>
> Finally, to still answer the reviewer’s question about “what” domains, we would like to share that we have run our algorithms on image datasets as well as document datasets, to name a few, and have observed interesting improvements that we are compiling for a separate project. We want to emphasize that we mention this just to answer the reviewer’s query about what other domains our algorithm could apply to, and this is not part of our contribution to this paper. We believe that our application on single-cell datasets is an important contribution and is the sole real-world experimental focus of this paper. The reviewer may ignore this paragraph if they seek more specific examples, as we do not wish to deviate any further from the paper's focus.
>
> Overall, if the reviewer has any other concerns about the paper itself, please let us know. We will try our best to answer them. Thank you for the discussion.

---

### Official Review · Reviewer_SH4R · 2024-11-04

**Soundness:** 2
**Presentation:** 3
**Contribution:** 3
**Rating:** 6
**Confidence:** 4

**Summary:**

The paper presents a new approach for achieving balanced rankings in graphs that have community structures. It addresses the problem of unbalanced rankings produced by traditional centrality measures. The authors introduce a structural concept called Multi-Core Periphery with Communities (MCPC), which combines both community and core-periphery structures. They propose "relative centrality" and develop a ranking algorithm that produces more balanced results than common centrality methods. The paper includes a theoretical analysis of ranking imbalances with MCPC structure and shows how their relative centrality approach resolves this issue. The paper demonstrates that their method improves clustering accuracy while achieving greater ranking balance compared to existing methods.

**Strengths:**

1. The paper introduces the concept of "relative centrality" and proposes a new structural assumption called Multi-Core Periphery with Communities (MCPC), which combines community structure and core-periphery structure.

2. The paper provides theoretical analysis of their proposed methods, including proofs of unbalancedness with MCPC structure and how their relative centrality approach overcomes this issue.

3. The paper demonstrates the usefulness of their balanced ranking algorithm on real-world data, specifically in improving the inference of community structure in single-cell RNA sequencing data.

4. The authors compare their method against several popular centrality measures and provide extensive simulations on real-world datasets.

**Weaknesses:**

1. The paper focuses primarily on directed graphs, which may limit the applicability of the methods to certain types of networks.

2. While the authors mention some existing work on multi-core structures, they don't provide a comprehensive comparison with these methods.

3. The paper briefly addresses the computational complexity of M-Rank for k-regular directed graphs in Section 3.2, but lacks analysis for other approaches, such as N2-Rank and RN-Rank. Providing additional clarification or a more comprehensive complexity analysis, especially for larger or irregular graphs, would enhance the paper's practical relevance for large-scale network applications.

4. Although the authors tested their method on 11 diverse single-cell datasets, these datasets are relatively small—only the TM dataset reaches 54K data points, with others below 16K. The superior results on Onion approach on the TM dataset in Table 2 raise questions about the scalability of the MCPC method on larger datasets. Besides, the PR for Onion is higher (.98) than RN-Rank (.87), yet RN-Rank is incorrectly highlighted in bold, which should be corrected. Evaluating the method on more larger datasets (e.g., millions of data points) could strengthen the paper’s contribution.

5. While the PR scores are high for RN-Rank and N2-Rank, the Purity metric is consistently lower than traditional centrality measures across most datasets in Table 2. The paper would benefit from a more in-depth discussion of this trade-off between Preservation Ratio and Purity, including potential ways to improve Purity scores while maintaining a high Preservation Ratio.

**Questions:**

1. How does the computational complexity of the proposed relative centrality algorithms compare to traditional centrality measures?

2. Can the MCPC structure and relative centrality concepts be extended to undirected graphs or weighted networks?

3. How does the performance of the proposed methods change as the number of communities in the graph increases?

4. How does the proposed method handle dynamic or temporal networks where the structure may change over time?

---

> ### Author Response · Authors · 2024-11-16
>
> We thank the reviewer for the thorough review. Please find our responses to the weaknesses mentioned and the questions asked below.
>
> ## *Responses to weaknesses:*
>
> **Focusing on directed graphs:** While we indeed are focused on directed graphs, this covers many applications, including analysis of graph embeddings of vector point clouds (not only in single-cell but also in other domains such as document and image datasets), which is a large and important area in Machine learning.
>
> ---
>
> **Comparison with other algorithms with multi-core structures:** As we have discussed in Appendix E2, most of the algorithms we mentioned focus on "discrete" detection of cores and peripheries focusing on undirected graphs, and furthermore, are generally quite slow (some of them taking $|V|^3$ time). We compared it with a prominent work focusing on multiple cores on directed graphs (and placed it in Appendix E2) and observed that the algorithm performs poorly in the MCPC structure of our interest.
>
> ---
>
> **Computational complexity:** We discuss the time complexity of the MR-Rank meta-algorithm in Section 3.2 and explicitly write them here. The time complexity or RN-Rank ($t$ steps ) on any graph with $G(V,E)$ is $ \mathcal{O}( |E|\cdot t)$ irrespective of the regularity of the graph. The analysis of N2-Rank is slightly more complex, with the exact expression being $ \mathcal{O}( (|E|\cdot t) + \sum_{u \in V} N_G(u))$. Overall, the runtimes are *almost linear* in the size of the graphs, which makes them scalable for large graphs. For example, our largest dataset has 54K points (around 810K edges). Here, our algorithm terminated in under 7 seconds on a Macbook M1 Air. Therefore, we think our algorithms should be able to handle datasets with millions of nodes. Our algorithms are also highly parallelizable, which can further improve the run time. Furthermore, the memory requirement of our algorithm is also linear in $|E|$, which is also a positive for scalability.
>
> ---
>
> **Concern on the TM dataset:** We thank the reviewer again for the thorough evaluation. We actually made a mistake in noting the preservation ratio for the TM dataset properly in Table 2. In this dataset, the "onion" method actually has the lowest preservation ratio, with PR $\approx 0.5$ for the top 0.2 fraction of points compared to $0.87$ for RN-Rank. We request the reviewer to look at Figure 19 in Appendix F, which provides the complete plots. It shows that as we select fewer points, the preservation ratios of the benchmark methods go down significantly. We will correct the mistake in the table in our revised version. To re-summarize, our methods have a higher preservation ratio across most of the datasets, irrespective of the graph size.
>
> ---
>
> **Purity vs preservation:** Improving purity as much as possible while maintaining a high preservation ratio is indeed a fundamental challenge. If we compare the tradeoffs, we can see that our improvement in purity is still comparable to the other methods, while our preservation ratios are significantly higher. However, we agree that one needs a better metric to unify these two scores, and we are working on it as a future step.
>
> ---
> ---
>
> ## *Answer to questions:*
>
> **Run-time compared to the traditional centrality measures:**  PageRank and degree centrality are probably the fastest centrality measures, boasting run-time of $\mathcal{O}(|E|)$. Even then, our method's runtime is comparable. In the aforementioned TM dataset, A sophisticated implementation of PageRank takes around 0.8 seconds, compared to approximately 6.8 seconds for a naive implementation of our algorithm. Here, we want to note that many other traditional centrality measures, such as Betweenness and closeness, have $\mathcal{O}(|E|^2)$ or even higher time complexity, making them impractical for large graphs.
>
> ---
>
> **Extension to undirected and weighted graphs:** The extension to weighted graphs is straightforward. In fact, our algorithm can be directly applied to weighted-directed graphs. We are in the process of acquiring a collection of natural undirected graphs with underlying communities to better understand what would be a useful formalism for MCPC structures in undirected graphs.
>
> ---
>
> **Performance dependence on the number of communities:** The runtimes of our algorithms do not depend on the number of underlying communities in the graph.
>
> ---
>
> **Handling of dynamic and/or temporal graphs:** This is an excellent question. Recall that our method consists of an initial centrality score (which involves a random walk) and then a local normalization procedure. Fast updation of random-walk-based procedures on dynamic graphs is an active area of research and should be treated as an independent problem. If the first step is robust to the change in the graph structure, then calculating the second step is relatively simple.
>
> ---
> ---
>
> We thank the reviewer again for the detailed discussion and will happily answer any other queries they may have.

---

> > ### Comment · Reviewer_SH4R · 2024-11-23
> > **Response to the Authors' Rebuttal**
> >
> > Thank you for the detailed response and for addressing most of the concerns.
> >
> > However, the primary issue is that the Purity metric is consistently lower than traditional centrality measures across most datasets in Table 2. The authors should suggest potential strategies or future directions to enhance Purity scores while maintaining a high Preservation Ratio.

---

> > > ### Author Response · Authors · 2024-11-23
> > >
> > > We are happy to hear that we have answered most of the concerns of the reviewer. Their remaining concern is that our method has lower purity than other ranking methods. Please find our response below.
> > >
> > > ---
> > >
> > > ### Purity-balancedness tradeoff:
> > >
> > > In this direction, we first point out that a *balancedness-purity* tradeoff can be inevitable in some cases. For example, consider that you have two underlying communities with an equal number of vertices (let's say $n$) that are very hard to separate. In such a case, if picking up some top fractions (such as 20%)  due to some ranking (PageRank) results in only one vertex being picked up from that community,  the purity will be very high ($1-5/n$). However, the preservation ratio and balancedness will be very low, which is really bad in our use cases (such as biological data). If the reviewer approves, we can add this intuition to the paper.
> > >
> > >
> > > *Traditional centrality measures can miss entire underlying communities:*
> > > Indeed, we notice that existing ranking methods such as PageRank sometimes entirely miss some hard-to-cluster communities or miss out on many points of the cluster. For example, this happens in the T-cell medicine dataset that we considered in the paper. This results in a higher purity (as the hard communities are missed to a significantly larger extent than our methods) in these methods.
> > >
> > > In contrast, our method selects points from all communities and is still able to generate a clustering that is comparable to the unbalanced ranking-based methods. In our rebuttal for the general audiences (https://openreview.net/forum?id=21rSeWJHPF&noteId=F6K1CehQFn), we have mentioned why balancedness and preservation ratio is very important in our application.
> > >
> > > ### Our focus is on quantifying and solving the unbalancedness issue:
> > >
> > > We re-emphasize that the goal of the paper is to address the unbalancedness of traditional centrality measures and design novel balanced ranking algorithms. As we have mentioned previously, we have succeeded in this (as visible from the PR values in Table 2), while maintaining comparable improvement in the separability.
> > >
> > > In fact, to the best of our knowledge, we are the **first paper** to both notice and systematically quantify and address this unbalancedness issue in traditional centrality measures and provide significant theoretically motivated improvement via our balanced ranking algorithms (that still identify the easier-to-separate cores) with practical applications.
> > >
> > > ---
> > > ---
> > > We hope our response gives the reviewer more context as to why wanting to maintain equal purity improvement while having improved balancedness may be unrealistic. To summarize,  the overall purpose of this paper is to address the unbalancedness issue while obtaining accuracy similar to that of existing methods. We believe that, in many cases, obtaining higher accuracy than our methods while maintaining similar balancedness may indeed be impossible. This is indeed an interesting (and primarily theoretically aligned) research direction. However, this lies completely out of the scope of the paper.
> > >
> > > We shall add a short discussion on the purity-unbalancedness tradeoff in the paper along the lines of our response above. We look forward to hearing the reviewer's response and shall try our best to alleviate their remaining concerns.

---

> > > > ### Author Response · Authors · 2024-11-24
> > > > **Further intuition on the inevitability of a purity-balancedness tradeoff**
> > > >
> > > > In our previous comment, we have described how there can be a purity-balancedness tradeoff. To further substantiate this intuition, we draw analogs from the fair-clustering paradigm, which is a supervised notion of balancedness in clustering (Each node has a provided group label, and one has to obtain the best clustering that maintains some proportionality of the number of nodes from any group in any of the output cluster). In such a case, it is currently being established that a quality-fairness tradeoff may be inevitable~[6].
> > > >
> > > > As we mentioned in our earlier comment, we aim to develop a quantitative analysis of the balancedness-purity tradeoff in the future for our unsupervised ranking problem. However, we do expect a tradeoff. We shall add this discussion to the paper.
> > > >
> > > > [6] ``The Fairness-Quality Trade-off in Clustering''. R. Hakim, A.A. Stoica, C.H. Papadimitriou, and M. Yannakakis, NeurIPS 2024.

---

> > > > > ### Comment · Reviewer_SH4R · 2024-11-25
> > > > >
> > > > > Thank you again for your response. I have reviewed all the reviews and answers, and I would like to maintain my current score.

---

> > > > > > ### Author Response · Authors · 2024-11-26
> > > > > >
> > > > > > We thank the revewier again for the feedback. We have a final question for the reviewer that we believe shall help us improve our exposition. The reviewer suggested we should further improve the purity score (which captures the accuracy of clustering outcomes), and it seems that they still consider this to be an issue.
> > > > > >
> > > > > > In response, we argued that balancedness-purity-improvement might be inevitable in some cases. We also gave a concrete example of how missing many points from one underlying community makes the purity score higher.
> > > > > >
> > > > > > In this direction, we would be thankful to hear the reviewer's opinion. Does the reviewer still think that both balancedness and purity can be improved arbitrarily? If so, we will try to develop better arguments to make the ``inevitability of tradeoff'' point clearer in the next round.
> > > > > >
> > > > > > If the reviewer also agrees that the tradeoff is inevitable, we would like to ask if the reviewer has further concerns that we can address.
> > > > > >
> > > > > > ---
> > > > > > ---
> > > > > >
> > > > > >
> > > > > > Additionally, we would also like to inform the reviewer that we have added a revised version of the paper with a new theoretical result (Theorem 3.3) that further provides proof of the balancedness of N-rank along with an improved representation. We thank the reviewer for their time and hope that the updated version can further help them judge our paper.

---

> > > > > > > ### Comment · Reviewer_SH4R · 2024-12-02
> > > > > > >
> > > > > > > Thank you for the detailed response. After thoroughly reviewing all reviewers' comments and the rebuttals, I have increased my score.

---

> > > > > > > > ### Author Response · Authors · 2024-12-02
> > > > > > > >
> > > > > > > > We thank the reviewer for a thorough and engaging discussion and are glad to hear they have decided to increase their score.

---

> ### Comment · Area_Chair_4J4d · 2024-11-25
>
> Could please acknowledge and respond to the rebuttal.

---

### Official Review · Reviewer_JR6C · 2024-11-08

**Soundness:** 3
**Presentation:** 3
**Contribution:** 2
**Rating:** 6
**Confidence:** 3

**Summary:**

The paper is motivated by the observation that traditional ranking algorithms can produce unbalanced rankings, and it aims to promote balancedness in centrality estimation. It first defines the concept of relative centrality and then proposes an iterative, graph-dependent local normalization of the centrality score. Empirical studies are provided to demonstrate the effectiveness of the proposed concepts.

**Strengths:**

S1. The paper aims to promote balancedness in nodes’ centrality ranking, using community detection as a concrete application scenario. I find this focus interesting.

S2. The paper proposes a multi-core-periphery structure with communities (MCPC) to quantify unbalancedness in centrality measures.

**Weaknesses:**

W1. The illustrative example in Figure 3 is unclear to me. The blue nodes in Figure 3(a) have more in-neighbors in Figure 3(b), but the out-degrees of the in-neighbors are also larger than those in Figure 3(b). Can we trivially conclude that the blue nodes in Figure 3(b) have smaller PageRank scores than those in Figure 3(a)?

W2. Following W1, if the answer is no, the core idea of defining the MCPC structure requires further clarification. Otherwise, the advantages of MCMC over traditional centrality measures (e.g., PageRank) seem marginal.

W3. The paper does not theoretically demonstrate the superiority of clusters detected by the proposed method over previous approaches, which limits the paper’s contributions.

------
The authors’ rebuttal partially addresses my concerns, and I would like to make a slight adjustment to my score.

**Questions:**

Could you provide further clarifications on W1 and W2 listed above?

---

> ### Author Response · Authors · 2024-11-16
>
> We thank the reviewer for their comments and questions. Please find our answers below.
>
> Please note that in Figures 3a and 3b, the out-degree of all vertices is the same (some edges are bidirectional). Therefore, in Figure 3b, the core nodes in the blue core will indeed have a lower score in PageRank compared to the core nodes in the red core, which causes unbalancedness (which we resolve with our relative centrality framework). We hope this resolves the reviewer's question.
> We had to use bidirectional edges to make the figure legible. We apologize for the confusion.
>
> Furthermore, we note that our formalism of MCPC structure not only allows us to capture the unbalancedness of traditional centrality measures but also provides a theoretical and simulation framework for designing a balanced ranking algorithm. This allows us to systematically develop simple (and therefore fast) ranking algorithms that can produce balanced ranking on several (in 10 of the 11 datasets that we tested) real-world datasets while ensuring the top points in the ranking are better separable into their underlying community. This further underlines the usefulness and importance of the MCPC structure formalism.
>
> We will be happy to answer any other questions the reviewer may have.

---

> > ### Comment · Reviewer_JR6C · 2024-11-17
> > **Response to the Authors' Rebuttal**
> >
> > Thank you for clarifying W1. Figure 3 is clear to me now.
> >
> > For W3, the paper theoretically demonstrates that the proposed method can produce a more balanced ranking result, which is good. However, the improvement in clustering results has not been theoretically analyzed. As a result, I am concerned whether we can trivially conclude that the proposed method improves clustering quality in general, rather than only in specific cases.

---

> ### Author Response · Authors · 2024-11-18
>
> We thank the reviewer for their engagement and appreciate that they liked our proof of balancedness. We would first like to highlight that we not only provide theoretical insight into our ranking being balanced but also prove that it has a high **core prioritization**. We prove this in Lemma A.7, which we refer to in Section 3.2. This implies that the top of the ranking consists of the *core* points from each community.
>
> Then, as a concrete application, we apply an existing graph clustering algorithm (Louvain) to the induced subgraph of the top-ranked vertices (cores) and see significant improvement in terms of clustering accuracy. This is primarily due to the fact that in the case of single-cell datasets, the cores are more separable; i.e., they have a **smaller fraction of inter-community edges**, as captured in the plots of Figures 6A (and Figures 10A to 20A). In Appendix F, we observe a similar phenomenon (cores being more separable) also happens in the graph embeddings of some generalizations of the famous Gaussian mixture model. We emphasize that we observe this higher separability of the top-ranked points and consequent improvement due to Louvain in **all** of the 11 single-cell datasets, indicating promising applications in future single-cell datasets. Kindly note that we do not claim any new clustering algorithm design in the paper. Instead, we simply apply Louvain on cores (top-ranked vertices).
>
>
> In conclusion, we re-emphasize that our focus is on designing a *balanced* ranking algorithm with *high core prioritization*, which we both theoretically support and experimentally justify. Such ranking algorithms can have other applications beyond clustering, depending on the significance of cores in different domains. Therefore, analysis of specific clustering algorithms and design of new clustering algorithms is not the focus of the paper, and our improvements in the clustering performance are due to the (more separable) core points of the communities being ranked at the top (in a balanced manner) by our ranking algorithms.
>
> We thank the reviewer again for the discussion and will be happy to answer any other questions they may have.

---

> > ### Author Response · Authors · 2024-12-02
> >
> > Dear reviewer,
> >
> > As the discussion period is nearing an end, we wanted to know if we could answer any other questions. We also wanted to let the reviewer know  (as we had mentioned in an earlier general audience comment) that we have significantly improved the presentation of the paper and added a new theorem that strengthens our balancedness results in our new version.
> >
> > We'll try our best to alleviate any other concerns the reviewer has about our paper. We thank them for their efforts.

---

> > > ### Comment · Area_Chair_4J4d · 2024-12-02
> > >
> > > Reviewer JR6C , could you please reply to the author's last post. Thanks.

---

> > > ### Comment · Reviewer_JR6C · 2024-12-03
> > > **Response to the rebuttal**
> > >
> > > Thank you for the further clarification! I have also read the authors’ responses to other reviews, where concerns about evaluating the superiority of the proposed metric are frequently mentioned. The authors have highlighted their contributions in a reasonable manner, and I appreciate their effort. As a result, I would like to slightly increase my score. However, I remain concerned about the applicability of the proposed metric, as its advantages in identifying clustering seem to be demonstrated only in relatively extreme scenarios.

---

> > > > ### Author Response · Authors · 2024-12-03
> > > >
> > > > We thank the reviewer for appreciating our efforts in presenting the contribution. As our setting is relatively new, we could not rely heavily on any established domain to present our results. For example, although centrality measures (such as PageRank) are a famous class of algorithms, we were surprised to notice that no systematic study of their unbalancedness exists.
> > > >
> > > > Regarding our metrics, we welcome any concrete suggestions from the reviewer. While we believe we were able to motivate and verify the need for balanced ranking in (a large set of) single-cell data, we will be very happy to explore suggestions the reviewer may have regarding these experiments that could further cement our contributions.
> > > >
> > > > We thank the reviewer again for their efforts.

---

### Author Response · Authors · 2024-11-16
**General audience comment**

We thank all reviewers for their comments. We observe there are some general questions about our motivation, our datasets, and the connections between our paper and the ICLR community, and we address them here.

---

### **The reason behind our focus on single-cell data and improvement of clustering:**

The motivation behind our project comes from our attempts to study single-cell RNA sequencing data (which we refer to as single-cell data), which is an emerging (and important) kind of genomics data.
While this motivated our initial ranking algorithm and framework, we believe it could be of independent interest as our algorithm does not use any domain knowledge of single-cell analysis. However, since our main focus is single-cell, we mainly use datasets from this domain to test our algorithm. For single-cell datasets, improvement in the performance of community detection (clustering) algorithms is an important objective, and therefore, we focused on this step. (Note that our theoretical model also combines core-periphery structure with ``community structure''). We agree with the Reviewer mu1B that the recommender systems is an application of interest that we shall explore in the future.

**TL;DR:** Though our motivations and consequent experiments mainly focus on single-cell data, we believe it can also be applied to different datasets as we did not use domain knowledge of single-cell data in our algorithm.

---

### **Background of single-cell analysis and its relevance to the ICLR community:**

Two reviewers pointed out that our sole focus on single-cell data was a weakness and requested a further explanation of the data. Here, we note that genomics analysis is an area that is gaining interest from the ML community. For example, ICLR 2024 hosted a workshop called *Machine Learning for Genomics Explorations (MLGenX)*. Besides, such workshops are frequently hosted by ICLR, NeurIPS, or ICML. Single-cell data analysis is a very important topic in genomics, providing a quantitative way to understand cells. The Science journal noted it as the *Breakthrough of the Year* in 2017.


Now, we quickly discuss the background of single-cell analysis. In single-cell data, we are given a dataset with some $n$ data points (each data point corresponding to a single cell), with $d$ features (each feature corresponding to the gene expressions of a cell). In single-cell analysis, the main goal is to understand cell behavior (understanding biological systems, diseases, and others) through gene expression.


Here, separating the data points into different clusters (according to their cell types) using gene expression is an important step in single-cell analysis, as noted by this popular Nature review paper [1]. Once the different communities are found, bioinformaticians then use it for different downstream tasks [2]. These downstream analyses have led to (and promise further advancements in) detecting genes responsible for different medical conditions and are being used to create new immunotherapy, among other applications. Therefore, better separating the data points into their underlying communities can lead to better performance of **all** of the downstream tasks, making it an impactful contribution.

**TL;DR:** Single-cell analysis has been gaining attention within the ML (including ICLR) community, and improving clustering performance is a fundamental ML problem.

---

[1] ``Best Practices for Single-cell Analysis across Modalities.'', Lukas Heumos et al., Nature Reviews Genetics, vol. 24, no. 8, 2023, pp. 550-572,  https://doi.org/10.1038/s41576-023-00586-w

[2] ``Single-nucleus Cross-tissue Molecular Reference Maps toward Understanding Disease Gene Function.'', Gökcen Eraslan et al., Science, 2022, https://www.science.org/doi/10.1126/science.abl4290

---

> ### Author Response · Authors · 2024-11-16
> **General audience comment (continued)**
>
> ###  **Usefulness of balanced ranking algorithm:**
>
> Finally, against this backdrop, we further motivate the usefulness of a balanced ranking algorithm, which is the primary algorithmic contribution of our paper.
>
>
> The state-of-the-art clustering algorithm for single-cell data is Seurat, which primarily consists of applying a graph clustering algorithm called Louvain on a KNN-like graph embedding of the data. Here, it is important to note that single-cell datasets are high-dimensional (usually more than 25,000) noisy data, suffering from technical noise and experimental error [2]. Many data points could be abnormal cells, such as dead cells and doublets, or cells affected adversely by experimental noise. Therefore, separating such cells in the graph embeddings may be very hard. In our MCPC structure, we capture them as peripheries.
>
> In our large-scale experiments, we observed that Louvain performs poorly on these peripheries, which inspired our ranking motivation: we should first rank and select the better separable cells (we capture them as cores in our MCPC structure) and then apply Louvain on these cores. In this context, balancedness and preservation are self-evidently desirable properties of a ranking algorithm as one hopes to select cores for each cell type.
>
>
> We believe that our experiments showed that we made good progress in this direction: the clustering performance can be significantly improved on cores while providing higher preservation compared to other ranking algorithms. On a more general level, ``balancedness'' is a property that seems quite natural for a ranking algorithm, and the current literature does not have such algorithms. Therefore, we propose our framework as a ranking algorithm, with the primary application explored in this paper being towards improving community detection. Furthermore, our ranking algorithm does not rely on domain knowledge of single-cell data. We believe it can be applied to high-dimensional noisy data from other domains.
>
>
> **TL;DR:** A balanced ranking algorithm based on relative centrality can improve the quality of popular clustering algorithms in this domain while still containing points from each underlying community, and such balancedness seems like a naturally desirable property when dealing with data with multiple cores.

---

### Author Response · Authors · 2024-11-26
**Revised paper**

We thank the reviewer for the discussion so far. Based on the overall discussion, we have revised our paper as follows.

1) In the initial review, several of the reviewers raised a question about our focus on single-cell data, which we addressed in a general audience comment above. We add these discussions to the paper in Section 2.

2) Restructuring and cleanup: We have changed the structure of the paper slightly following the recommendations of reviewer 6NAW. We have moved the "Related works" section to the end of the introduction and the definitions of MCPC structure to Section 2, leaving Section 3 to completely focus on explorations in the random graph model and our algorithm design.

3) Stronger theoretical support: In the first round of review, some reviewers (JR6C and SH4R) liked our theoretical result. In contrast, reviewer 6NAW commented that our theoretical support for the balancedness of N-Rank was not very strong. Initially, we had shown that the score of points from both cores will be high ($1-o(1)$), whereas in degree centrality, the score of a core vertex depends on the concentration of the core.

*New theorem:* In the revised version, in Theorem 3.3, we explicitly show that $\theta(1/k)$ fraction of points from each core will be given a score of $1$ (highest possible value) by N-Rank in expectation. The proof can be found in Appendix A.4. This provides further theoretical support for the balancedness of relative centrality. We also highlight the limitations of our current theoretical analysis and future direction.

4) Reviewer SH4R had a remaining doubt (they said we had answered most of their original queries) about the points selected by our method, leading to weaker clustering improvement compared to the traditional centrality measures. We believe we answered this question rigorously by pointing out that if the points selected are highly unbalanced, then one may trivially get a high clustering accuracy. However, this unbalancedness can be crucial, as it removes information about complete clusters from the dataset (this indeed happens with the traditional centrality measure). We further pointed out that our method still shows comparable improvement in the clustering accuracy while having significantly higher balancedness. We also pointed out that similar quality-fairness tradeoffs also exist in the "supervised" fair-clustering problem. We have added this discussion to Appendix E.

---
---
To conclude, the main changes are:

1) Adding more discussion on single-cell data so that readers can appreciate its importance.
2) Adding new Theorem 3.3 (proof in Appendix A.4)
3) Restructuring Section 3 to move the definitions of MCPC to Section 2
4) A small discussion on the inevitability of clustering improvement-balancedness tradeoff is added in Appendix E.

---

### Meta-Review · Area_Chair_4J4d · 2024-12-19

**Metareview:**

This paper addresses the issue of unbalanced rankings in graph structures with underlying communities, a limitation of traditional centrality-based ranking algorithms like PageRank. By introducing a new method called relative centrality, the authors promote more balanced rankings, validated through theoretical analysis, simulations, and applications to single-cell data, improving clusterability and community representation.

The paper has been well received by 3/4 reviewers and the authors addressed their concerns during the rebuttal period. Reviewer mu1B raised a concern about the narrow applicability of the new centrality approach to a specific type of datasets. I agree with the reviewer that there are a lot of centrality methods. However, based on other reviewer's comments, the authors seem to clearly demonstrate a case where their approach is better.

**Additional Comments On Reviewer Discussion:**

See my comments above.

---

### Decision · Program_Chairs · 2025-01-22

Accept (Poster)